# SEPepQuant enhances the detection of possible isoform regulations in shotgun proteomics

Yongchao Dou[1,2], Yuejia Liu[3], Xinpei Yi[1,2], Lindsey K. Olsen[1,2], Hongwen Zhu[4], Qiang Gao[5], Hu Zhou [3,4] & Bing Zhang [1,2] ✉

Shotgun proteomics is essential for protein identification and quantification in biomedical research, but protein isoform characterization is challenging due to the extensive number of peptides shared across proteins, hindering our understanding of protein isoform regulation and their roles in normal and disease biology. We systematically assess the challenge and opportunities of shotgun proteomics-based protein isoform characterization using in silico and experimental data, and then present SEPepQuant, a graph theory-based approach to maximize isoform characterization. Using published data from one induced pluripotent stem cell study and two human hepatocellular carcinoma studies, we demonstrate the ability of SEPepQuant in addressing the key limitations of existing methods, providing more comprehensive isoform-level characterization, identifying hundreds of isoform-level regulation events, and facilitating streamlined cross-study comparisons. Our analysis provides solid evidence to support a widespread role of protein isoform regulation in normal and disease processes, and SEPepQuant has broad applications to biological and translational research.

Alternative splicing of precursor messenger RNA (pre-mRNA) is an essential post-transcriptional process that is believed to underlie increased cellular and functional complexity in eukaryotic organisms[1,2]. This process is highly regulated, and dysregulated RNA splicing has been linked to a wide range of diseases such as retinal and developmental disorders, neurodegenerative diseases, and cancer[3,4]. High-throughput sequencing-based transcriptomic studies have shown that most human protein-coding genes undergo alternative splicing to produce multiple mRNA isoforms[5]. Mass spectrometry (MS)-based shotgun proteomics is the primary method for protein identification and quantification from biological samples[6], but shotgun proteomics studies have provided very limited information on protein

isoforms due to intrinsic challenges in data analysis. In fact, the extent to which transcript isoform complexity propagates to the proteome remains controversial[7–9], and systematic investigation of the roles of protein isoforms in normal and disease biology is largely lacking[10].

In a shotgun proteomics experiment, proteins extracted from biological samples are digested into peptides using enzymes such as trypsin and then analyzed by liquid chromatography-tandem mass spectrometry (LC-MS/MS). Each LC-MS/MS run generates thousands of spectra, which serve as the basis for the identification and quantification of peptides and proteins. Many bioinformatics tools have been developed to perform these essential computational tasks in shotgun proteomics data analysis, such as MaxQuant[11], Trans-proteomic

[1]Lester and Sue Smith Breast Center, Baylor College of Medicine, Houston, TX 77030, USA. [2]Department of Molecular and Human Genetics, Baylor College of Medicine, Houston, TX 77030, USA. [3]School of Chinese Materia Medica, Nanjing University of Chinese Medicine, 138 Xianlin Avenue, 210023 Nanjing, Jiangsu, China. [4]Department of Analytical Chemistry, State Key Laboratory of Drug Research and CAS Key Laboratory of Receptor Research, Shanghai Institute of Materia Medica, Chinese Academy of Sciences, 555 Zuchongzhi Road, 201203 Shanghai, China. [5]Department of Liver Surgery and Transplantation, Liver Cancer Institute, Zhongshan Hospital, Fudan University, and Key Laboratory of Carcinogenesis and Cancer Invasion of Ministry of Education, 180 Fenglin Road, 200032 Shanghai, China. ✉e-mail: bing.zhang@bcm.edu

pipeline[12], OpenMS[13], FragPipe[14,15], among others. Despite algorithmic and implementation differences, these tools share a similar workflow. First, observed MS/MS spectra are searched against a reference protein database for peptide identification. Next, identified peptides are used to infer a list of proteins that are assumed to be present in the sample, a process known as protein inference. Finally, inferred proteins are quantified based on the signal intensity measurements of the constituting peptides.

The difficulty in protein isoform identification and quantification is complicated by the large number of degenerate peptides, which are peptides that can be mapped to multiple proteins due to a high level of sequence similarity between protein isoforms encoded by the same gene, or genes in the same gene family. The current most adopted practice in the field is to collapse proteins with the same set of supporting peptides together with those that are supported by a subset of these peptides into a protein group[16,17]. For protein quantification, peptides shared by multiple protein groups are either ignored or assigned to the group with the largest number of associated peptides, i.e., defined as razor peptides by MaxQuant[11]. Typically, one representative protein (i.e., the one with the largest number of associated peptides) is selected from each protein group for reporting and downstream analysis[14,18]. This parsimonious approach plays a critical role in preventing overstating the number of proteins in protein inference, however, it also limits the potential for protein isoform characterization. First, protein isoforms without uniquely identified peptides are essentially ignored. Secondly, the assignment of shared peptides to the protein groups and proteins with the largest number of associated peptides for quantification may not necessarily be the correct solution.

To address the challenge in protein isoform discrimination, several methods have been developed. One solution is to perform gene-based quantification, which is implemented in tools such as gpGrouper[19] and FragPipe and used in some recent studies[20–22]. gpGrouper uses quantities of gene-specific peptides to guide the split of quantities of shared peptides. Because only a small proportion of peptides are shared between genes, it makes full use of both unique and shared peptides to produce gene-level quantification with demonstrated accuracy[19]. However, protein isoform information is ignored in this approach. Along the same line, SCAMPI uses statistical modeling to generate quantification for individual proteins using both unique and shared peptides[23]. This approach was demonstrated when proteomics data were searched against the UniProt canonical database in which only canonical protein sequences are included, which means a single canonical sequence for most genes. For example, although there are eight annotated isoforms of TP53 in UniProtKB, the canonical database only included a single canonical isoform. Therefore, isoform reduction occurred in database construction, and the challenge addressed was primarily to distribute quantities of peptides shared by different genes. The method's ability to handle search results from a comprehensive protein database is uncertain because the inclusion of canonical and alternative isoform sequences leads to a substantial reduction in the number of isoform-specific peptides, which are crucial for accurately distributing the quantities of peptides shared between isoforms. Moreover, it is also unclear whether the method can be directly applied to ratio data generated from labeled experiments such as tandem mass tag (TMT)-base experiments.

Methods have also been developed based on the assumption that the quantitative pattern of peptides derived from one protein will correlate over several samples. Protein Quantification and Peptide Quality Control (PQPQ) selects peptides correlating over samples to improve the quantitative accuracy and precision[24], whereas Peptide Correlation Analysis (PeCorA) focuses on outlier peptides to reveal differential proteoform regulation[25]. These methods require multiple samples for analysis, and peptides annotated to more than one protein are excluded from the analysis. Therefore, many peptides would be excluded when a comprehensive protein database covering both canonical and alternative isoform sequences are used for database searching.

Leveraging RNASeq data from matched samples, the Custom-ProDB approach constructs a customized protein database with reduced number of protein isoforms by excluding isoforms with low transcript abundance in matched RNASeq data[26]. This method also enables novel protein isoform identification, but the quantification challenge remains because each gene may still have multiple RNASeq data-supported isoforms. Based on a strong assumption that each gene only has a dominant isoform, Liu et al.[10] used matched RNASeq data to select one isoform with the highest transcript abundance for each gene. Woo et al.[27] and Lau et al.[28] focus on novel protein isoforms by identifying and quantifying peptides mapped to novel isoform junctions detected based on RNASeq data. While these methods are appealing, their utility in the majority of proteomic studies is limited by the prerequisite of matched RNASeq data.

In this study, we systematically assess the challenge and opportunities of shotgun proteomics-based protein isoform characterization using in silico digestion data and experimental data from a published induced pluripotent stem cell (iPSC) study[28] and two published human hepatocellular carcinoma (HCC) studies[29,30]. To tackle the challenge of protein isoform characterization and leverage the potential opportunities, we extend the bipartite graph representation of peptide-protein relationships[17] to a tripartite graph for a comprehensive representation of the peptide, protein, and gene relationships. From the tripartite graph, we define a new quantification unit called Structurally Equivalent PEPtides (SEPEPs). These SEPEPs consist of peptide vertices that are connected to precisely the same set of protein vertices within the graph and are thus structurally equivalent in the graph. To facilitate downstream interpretation, we further divide the SEPEPs into five classes based on their patterns of connections to source proteins and genes in the tripartite graph. The introduction of SEPEPs as the quantification unit represents a significant innovation. It fundamentally differs from existing quantification approaches that employ parsimonious protein groups, individual genes, individual proteins, or correlated peptides from individual proteins, as the unit of quantification. While using peptides mapping exclusively to a single protein for quantifying that specific protein, as implemented in PQPQ and PeCorA, provides accurate quantification, it excludes many peptides that are shared by multiple proteins. On the other hand, when parsimonious protein groups or genes are employed as the quantification units, the isoform-specific information available from shotgun proteomics data is often suppressed or lost. By shifting the quantification unit to peptides that exclusively map to a group of protein isoforms that are indistinguishable based on the identified peptides, our SEPEP-based method can leverage all confidently identified peptides, including those mapping to multiple proteins or even multiple genes. Moreover, this approach retains and utilizes all the available isoform-distinguishable information present in the data, thus enhancing the detection of possible isoform regulations in our analysis. Using the iPSC and HCC datasets, we demonstrate the ability of our approach in addressing the key limitations of the parsimony-based methods, providing more comprehensive proteome characterization, identifying hundreds of isoform-level regulation events, and enabling streamlined cross-study comparisons.

## Results
### Assessing the challenge and opportunities of isoform characterization
Among the 19,449 protein-coding genes annotated in the RefSeq database, 14,698 (75.6%) have more than one protein isoforms, and 3409 (17.5%) have 10 or more protein isoforms (Fig. 1a). Most isoforms from the same gene have very high sequence similarity (>90%, Fig. 1b), highlighting the challenge in discriminating isoforms in shotgun

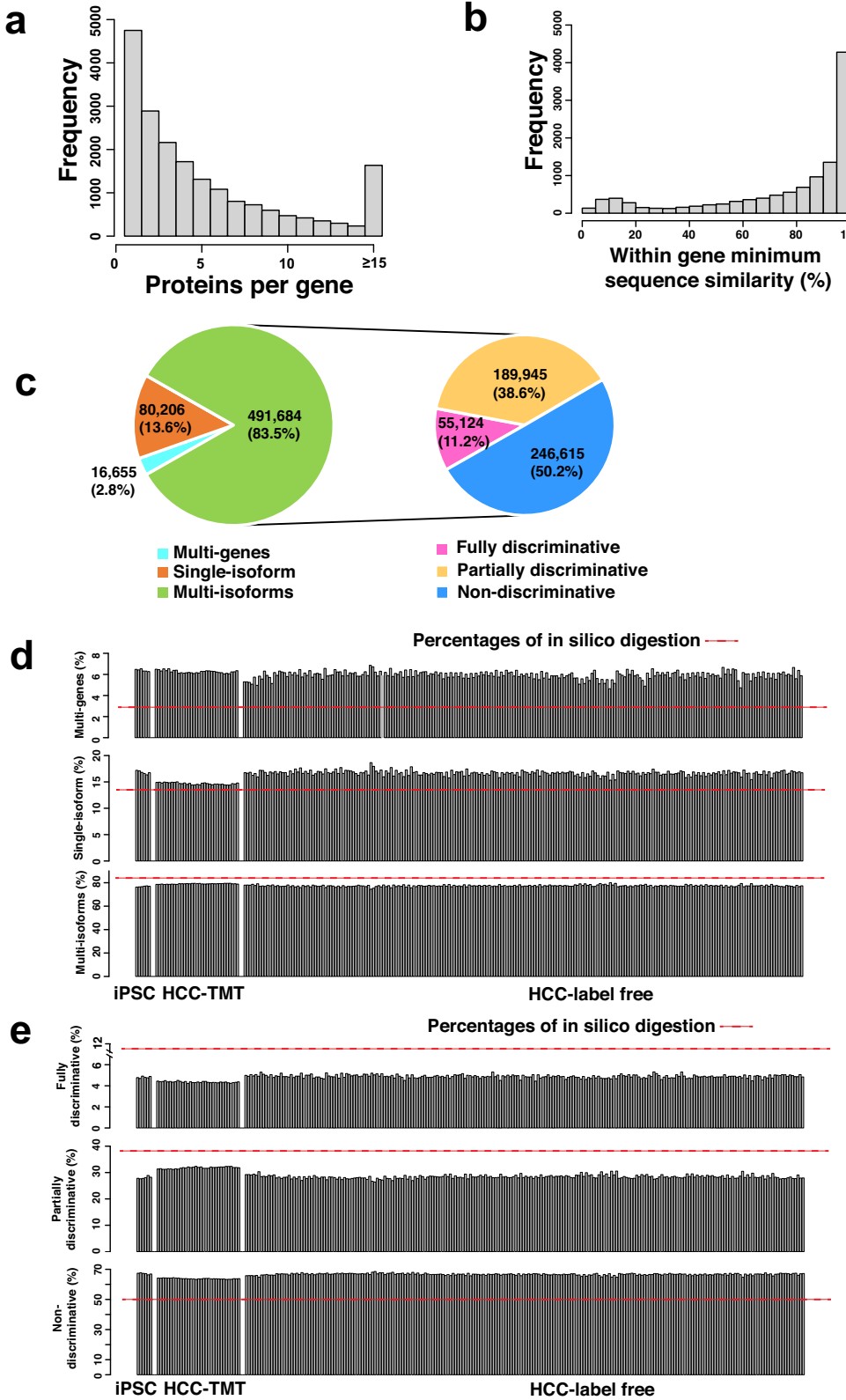

**Fig. 1 | Assessing the challenge and opportunities of protein isoform characterization.** a Distribution of the number of protein isoforms for all coding genes. **b** Distribution of the minimum within-gene protein isoform sequence similarity. **c** Classification of the in silico digested peptides based on their mapping to genes and protein isoforms. **d** Classification of the experimentally identified peptides in individual samples in an iPSC cell line study and two hepatocellular carcinoma (HCC) studies into three categories, equivalent to the pie chart on the left in (**c**). **e** Further classification of peptides in the multi-isoform group in d into three categories, equivalent to the pie chart on the right in (**c**). Source data of (**a**, **b**, **d**, **e**) are provided as a Source Data file.

proteomics experiments. However, among the 11,809 genes with three or more protein isoforms, 6165 (52.2%) have at least one pair of isoforms with a sequence similarity lower than 90%, or an average of one amino acid difference in every 10 amino acids, suggesting the possibility to identify isoform-discriminating peptide sequences for a substantial number of genes.

To further assess the challenge and opportunities of isoform characterization using shotgun proteomics, we performed in silico trypsin digestion of the RefSeq protein database to generate fully tryptic peptides with length 7 to 50 and with 0, 1, or 2 missed cleavages, or with semi-tryptic peptides (Fig. 1c and Supplementary Fig. 1). Taking peptide without missed cleavage as an example, among the 588,545 resulting peptide sequences, 2.8% could be associated to multiple genes (i.e., multi-gene peptides), 13.6% to genes with a single protein isoform (i.e., single isoform peptides), and 83.5% to genes with more than one isoform (i.e., multi-isoform peptides). Within the group of multi-isoform peptides, around half could be mapped to all protein isoforms of a gene, and thus providing no information for isoform discrimination (i.e., non-discriminative peptides); however, another half, or 246,615 peptides, could be uniquely mapped to one isoform (i.e., fully discriminative peptides) or a subset of isoforms (i.e., partially discriminative peptides) (Fig. 1c). Peptide distributions from other in silico digestion experiments were similar to that of the experiment with fully digested peptides with no missed cleavages (Supplementary Fig. 1).

Next, we compared our in silico digestion results with experimental data from a TMT-based iPSC study[28] and two human HCC studies[29,30], one TMT-based (HCC-TMT) and one label-free (HCC-label-free). Although two missed cleavage sites were allowed in database searching, less than 5% of identified peptides had missed cleavage sites (Supplementary Dataset 1). Around 6% of the peptides identified in these studies were multi-gene peptides, and the ratios more than doubled the 2.8% estimate from the in silico digestion (Fig. 1d). This observation may be explained by the higher likelihood of detecting these peptides in data-dependent MS experiments because they can be derived from multiple genes. Percentages of the single isoform peptides in these studies were slightly higher than that in in silico digestion, whereas an opposite trend was found for multi-isoform peptides (Fig. 1d). Within the group of multi-isoform peptides, the ratios of non-discriminative peptides were about 15% higher in these studies than those in in silico digestion, likely due to contributions from multiple isoforms (Fig. 1e). Percentages of the fully discriminative peptides and partially discriminative peptides in these studies were lower than those in in silico digestion, however, they still accounted for about 35% of the multi-isoform peptides, or about 27% of all identified peptides (Fig. 1e).

In summary, experimental data are largely consistent with the in silico digestion results, and both suggest that despite intrinsic challenges, there are a substantial fraction of peptides that hold important information for isoform characterization in shotgun proteomics.

## Tripartite graph modeling of peptides identified by shotgun proteomics

Peptides shared by multiple genes or multiple protein isoforms of the same gene complicate protein inference and quantification. Based on Occam's razor or the principle of parsimony, the current best practice in the proteomics field is to collapse proteins with the same or subset of supporting peptides into a minimal list of protein groups, and for quantitative rollup, peptides shared by multiple proteins are assigned only to the ones with the most identification evidence. Although practically useful, this parsimonious approach greatly limits the potential for protein isoform characterization. We propose a tripartite graph modeling approach to represent the data more accurately.

The tripartite graph modeling approach involves four major steps (Fig. 2a–d). First, a tripartite graph is built with three sets of vertices representing all peptides identified in a study (Pep1-Pep12), proteins to

which the peptides can be mapped (Pro1.1–Pro3.1), and host genes of the proteins (Gene1–Gene3), respectively, and the vertices are connected by edges indicating their mapping relationships (Fig. 2a). Second, using a graph theory-based technique[31,32], peptides connected to exactly the same set of protein vertices are grouped together and defined as a group of structurally equivalent peptides (SEPEP), leading to eight SEPEPs in Fig. 2b. To clarify, the term SEPEP is used to denote a specific grouping of peptides that exhibit structural equivalence within the context of the tripartite graph instead of an individual peptide. Of note, the gene vertices do not affect the identification of SEPEPs, but they help organize protein vertices into genes and classify SEPEPs into single-gene or multi-gene SEPEPs (see below) to facilitate data interpretation. Third, the target-decoy approach[33] is used to estimate false discovery rate (FDR) at the SEPEP level. Specifically, a SEPEP is considered a target hit if the peptides in the SEPEP are from a forward protein sequence, and a decoy hit if the peptides are from a decoy protein sequence. SEPEPs with FDR >0.01 are excluded from further analysis (Fig. 1c). Finally, the remaining SEPEPs are classified into five classes based on their patterns of connections to source proteins and genes in the tripartite graph (Fig. 2d). Class 1 through 5 correspond to single isoform SEPEPs, fully discriminative SEPEPs, partially discriminative SEPEPs, non-discriminative SEPEPs, and multi-gene SEPEPs, respectively. Class 1 to 4 SEPEPs are labeled with gene name followed by SEPEP order within the gene and SEPEP class type, e.g., Gene1_SEPEP.1_C1. Class 5 SEPEPs are labeled with "Multiple" followed by SEPEP order across the whole study and C5, e.g., Multiple_SEPEP.1_C5. Each SEPEP is also identified by associated gene(s) and protein(s), both alphabetically sorted and concatenated, in a mapping table, which will allow streamlined cross-study comparison.

In contrast to existing methods that make protein inference and then use protein groups or gene groups as the reporting and quantification units, our approach uses SEPEPs as the reporting and quantification units (Fig. 2e). All methods share the same database searching, peptide-spectrum match (PSM) FDR control, and peptide FDR control protocols. SEPEP identification and SEPEP-level FDR control are performed in parallel to the standard protein inference and protein level FDR control. Finally, the same algorithm can be used to report quantification at SEPEP, gene group, and protein group levels. Specifically, each quantification unit (protein group, gene group, or SEPEP) is quantified based on abundance of all associated peptides using an appropriate method, such as mean, median, or sum, according to the nature of the proteomics experiment. SEPEP-based quantification is referred to as SEPepQuant.

## SEPepQuant enables more comprehensive proteome characterization

We applied SEPepQuant to the iPSC, HCC-TMT, and HCC-label-free datasets mentioned above, using FragPipe for database searching. The numbers of identified SEPEPs ranged from about 10,000–25,000 for each sample or TMT plex, driving by the depth of the proteomics studies (Fig. 3a). Although the numbers of identified SEPEPs varied greatly across different datasets, the percentages of SEPEPs failing 1% FDR filtering were around 15% for most samples, except for 11 samples from the HCC-label-free dataset (Supplementary Fig. 2a). About 90% of the rejected SEPEPs in these datasets contained only a single peptide with a mean peptide number of about 1.04 (Supplementary Fig. 2b). Therefore, the rejected SEPEPs may represent less robust identifications. After quality control, there were still about 500–3500 class 5 SEPEPs (Fig. 3b). These multi-gene peptides are typically removed or assigned to a single gene with the strongest identification evidence in parsimonious protein inference, leading to information loss or potential misinterpretation.

The SEPepQuant results were further filtered by removing SEPEPs with more than 50% missing values in each dataset. SEPEPs passing this criterion were considered as quantifiable SEPEPs. The same criterion

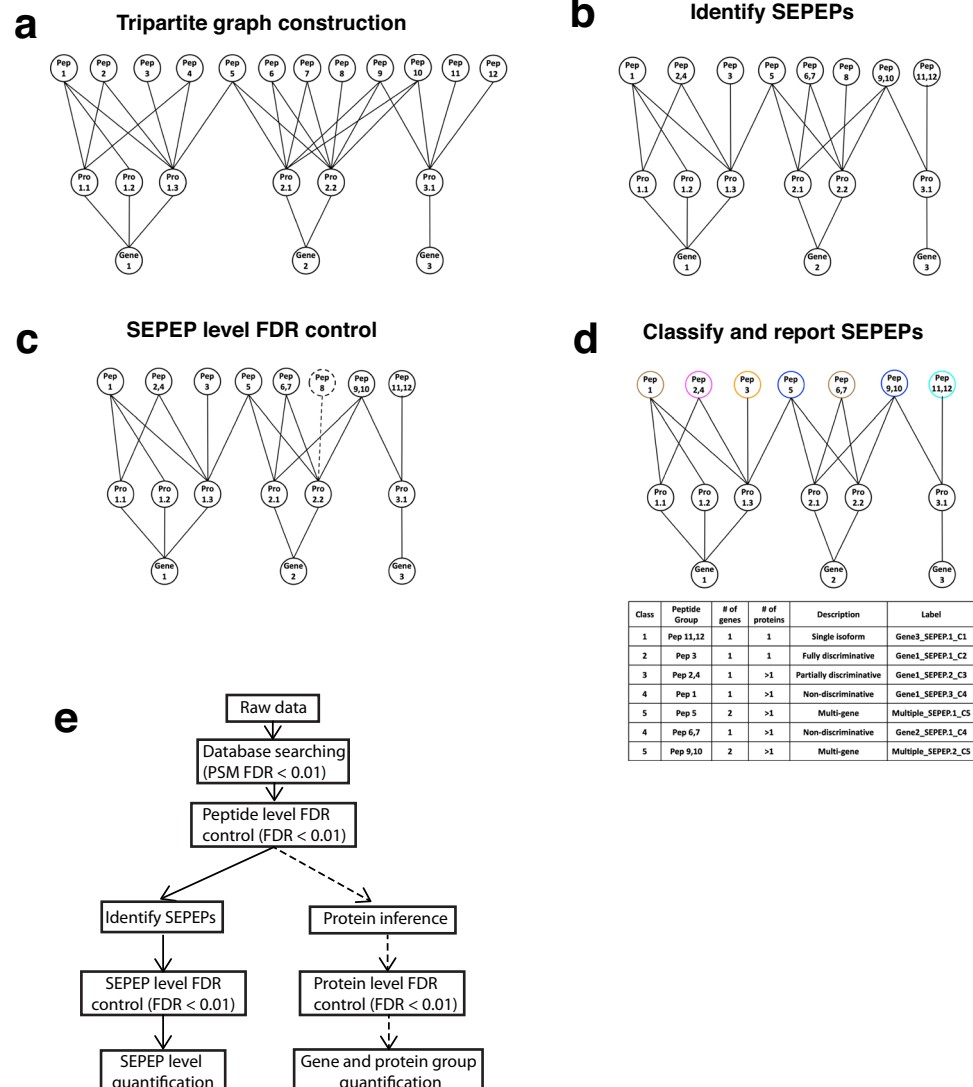

**Fig. 2 | Overview of the tripartite graph modeling approach. a** Tripartite graph construction: connect peptides to all proteins that contain them, and proteins to the host genes, to form a tripartite graph. **b** SEPEP identification: identify and group structurally equivalent peptides (SEPEP), i.e., peptides connecting to the same set of proteins in the graph. **c** FDR control: calculate SEPEP-level FDR and remove

SEPEPs with an FDR higher than the pre-specified threshold, e.g., FDR > 0.01. **d** SEPEP classification and reporting: classify SEPEPs based on their patterns of connections and report SEPEP-level quantification. **e** A comparison between SEPEP analysis procedure and the classical parsimony protein inference-based procedure.

was also applied to gene and protein group-level quantification derived from parsimonious protein inference. The total number of quantifiable single-gene SEPEPs were 8322, 17,543, and 7745 from the iPSC, HCC-TMT, and HCC-label-free datasets, respectively, and they corresponded to 5147, 8865, and 4936 genes (Fig. 3c and Supplementary Datasets 2–10). Compared with protein group-level quantification reported by FragPipe, which may also provide abundance of multiple distinguishable protein groups for individual genes, SEPep-Quant reported 5.8–33.8 time more genes with multiple features (Fig. 3d). To assess the broader applicability of this finding, we performed a supplementary analysis on the iPSC dataset using MaxQuant for database searching. Similarly, compared with the protein group-level quantification reported by MaxQuant, SEPepQuant identified 43 time more genes with multiple features (Supplementary Fig. 2c). We further compared genes harboring quantifiable single-gene SEPEPs with quantifiable genes reported based on parsimonious inference (Fig. 3e and Supplementary Datasets 2–10). The numbers of genes harboring quantifiable single-gene SEPEPs were smaller than corresponding gene numbers reported by parsimonious inference across all

three datasets. However, SEPepQuant also reported 1318, 3122, and 975 multi-gene SEPEPs for the three datasets, respectively, and such information is missing or difficult to track in existing computational tools (Fig. 3c).

Among genes quantified by both methods, the quantifications of the C4, non-discriminative SEPEPs exhibited a strong correlation with their respective host gene quantifications. Only a small fraction of cases, specifically 8% in the iPSC dataset, 1% in the HCC-TMT dataset, and 5% in the HCC-label-free dataset, showed correlations below 0.5 (Fig. 3f, black curves), emphasizing the reliability of SEPEP quantifications. Interestingly, for genes with multiple SEPEPs, a significant number had at least one SEPEP with a correlation less than 0.5 with their host genes, including 65.4%, 35.1%, and 79.8% of genes in the iPSC, HCC-TMT, and HCC-label-free datasets, respectively (Fig. 3f, pink curves). To further explore this observation, we assessed the distribution of within-SEPEP peptide correlations and their relationship with average MS1 peptide intensities in the HCC-TMT dataset. As expected, peptides with higher MS1 intensities exhibited relatively higher correlations, but the overall correlation remained strong, with a

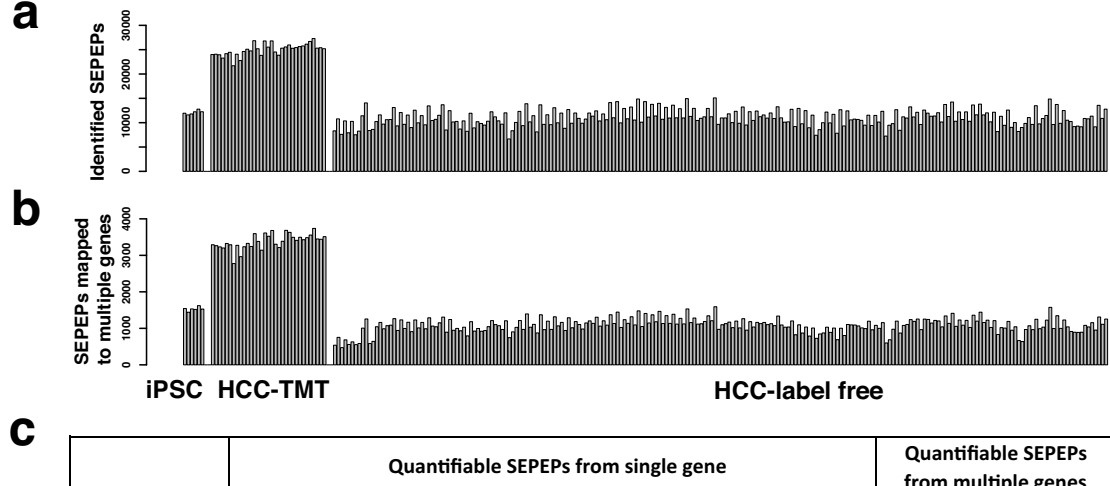

**a** Identified SEPEPs

**b** SEPEPs mapped to multiple genes

iPSC   HCC-TMT                    HCC-label free

**c**

| | Quantifiable SEPEPs from single gene | | | | | Quantifiable SEPEPs from multiple genes |
|---|---|---|---|---|---|---|
| Dataset | C1 | C2 | C3 | C4 | Total | # of genes | C5 |
| iPSC | 929 | 583 | 3369 | 3441 | 8322 | 5147 | 1318 |
| HCC TMT | 1531 | 1136 | 8661 | 6215 | 17543 | 8865 | 3122 |
| HCC label free | 907 | 516 | 3081 | 3241 | 7745 | 4936 | 975 |

**d**

| | | iPSC | | HCC-TMT | | HCC-label free | |
|---|---|---|---|---|---|---|---|
| | | SEPEP-based | Protein group | SEPEP-based | Protein group | SEPEP-based | Protein group |
| | Gene with multiple features | 2465 | 73 | 4892 | 219 | 2226 | 381 |
| Feature distribution | 2 | 1566 | 64 | 2410 | 186 | 1415 | 291 |
| | 3 | 552 | 4 | 1253 | 25 | 499 | 79 |
| | 4 | 221 | 3 | 630 | 4 | 181 | 7 |
| | 5 | 78 | 1 | 308 | 2 | 78 | 1 |
| | >=6 | 48 | 1 | 291 | 2 | 53 | 7 |

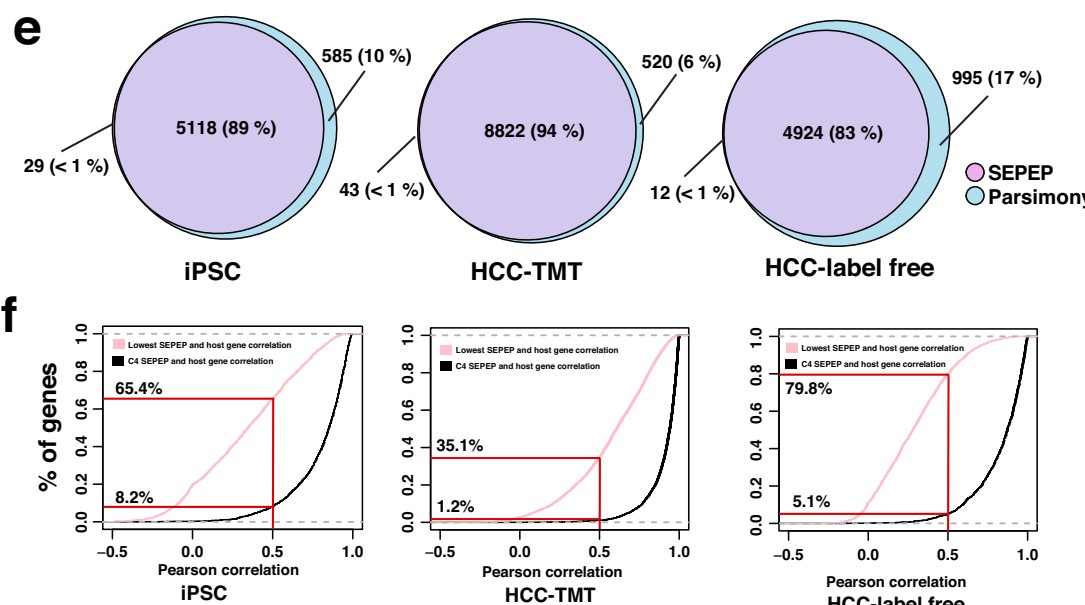

**e**

585 (10 %)

5118 (89 %)

29 (< 1 %)

iPSC

520 (6 %)

8822 (94 %)

43 (< 1 %)

HCC-TMT

995 (17 %)

4924 (83 %)

12 (< 1 %)

HCC-label free

● SEPEP
● Parsimony

**f**

65.4%

8.2%

iPSC

35.1%

1.2%

HCC-TMT

79.8%

5.1%

HCC-label free

% of genes — Pearson correlation

Lowest SEPEP and host gene correlation
C4 SEPEP and host gene correlation

**Fig. 3 | SEPEP-level quality control and quantification in three selected datasets. a** Numbers of identified SEPEPs in three studies. **b** Numbers of multi-gene SEPEPs passing FDR control. **c** Class distribution of the quantifiable SEPEPs. **d** Comparison of numbers of genes with multiple SEPEPs in SEPepQuant analysis and those with multiple protein groups in parsimonious inference. **e** Overlap of genes with quantifiable single-gene SEPEPs and quantifiable genes by parsimonious inference. **f** Distributions of the correlations between C4 SEPEPs and host genes (black curves) and the lowest correlations between SEPEPs and host genes (pink curves). Source data of (**a**, **b**, **e**, **f**) are provided as a Source Data file.

median value of 0.69 (Supplementary Fig. 2d). These findings suggest that at least some of the discordance between SEPEP and gene correlations could be attributed to isoform-specific regulation. Together, our results show that SEPepQuant provides additional resolution that may enable more comprehensive proteome characterization than traditional protein group-level or gene-level analysis.

## SEPepQuant addresses key limitations of the parsimony-based methods

In parsimonious protein inference, protein isoforms without uniquely identified peptides are largely ignored, and shared peptides are assigned to proteins with the most identification evidence for quantification. Here we selected representative examples from the HCC-TMT dataset to illustrate the drawback of these simplifications on protein isoform characterization, and the effectiveness of SEPepQuant in addressing these limitations.

*ACTR3* encodes two protein isoforms. The short isoform NP_001264069.1 is translated from a downstream translation initiation site and is shorter at the N-terminus compared to the long isoform NP_005712.1. Among the 17 peptides identified for this gene, three were unique to the long isoform, and others were shared between the two isoforms (Fig. 4a). Parsimonious protein inference assigned all shared peptides to the long isoform, and the quantification based on all peptides showed that NP_005712.1 was significantly lower in tumors compared with normal adjacent tissues (NATs) (*P* = 1.93e-4). Meanwhile, no quantitative information was provided for the short isoform. In contrast, SEPepQuant reported two SEPEPs. ACTR3_SEPEP.2_C2 was associated with the three NP_005712.1-specific peptides, and it was significantly higher in tumors compared with NATs (*P* = 2.30e-11). ACTR3_SEPEP.1_C4 was associated with the shared peptides, and it was lower in tumors compared with NATs with a marginal significance (*P* = 0.02). Despite the lack of NP_001264069.1-specific peptides, it was not difficult to infer that this isoform was suppressed in tumors based on the strong elevation of NP_005712.1. Thus, SEPepQuant provided useful information for both isoforms, whereas the parsimonious inference assigned all peptides to the long isoform, leading to inaccurate quantification because some peptides may actually belong to the short isoform despite being mappable to both isoforms.

*FKBP11* encodes three protein isoforms, and none of the six peptides identified for this gene could be uniquely mapped to a single isoform (Fig. 4b). Because NP_057678.1 could explain all six observed peptides but the other two isoforms could not, parsimonious protein inference assigned all six peptides to NP_057678.1, and the resulting quantification showed that this isoform was significantly increased in tumor samples compared with NATs. SEPepQuant reported quantification results for three SEPEPs. Both the non-discriminative FKBP11_SEPEP.1_C4 (one peptide) and the partially discriminative FKBP11_SEPEP.3_C3 (NP_001137253.1 and NP_057678.1, one peptide) had higher abundance in tumors, whereas the partially discriminative FKBP11_SEPEP.2_C3 (NP_001137254.1 and NP_057678.1, four peptides) had lower abundance in tumors. Although the change of NP_057678.1 remained difficult to determine, SEPepQuant results clearly suggested decreased abundance of NP 001137254.1 and increased abundance of NP_001137253.1 in tumor samples.

In addition to providing higher resolution for characterizing multiple protein isoforms encoded by the same gene, SEPepQuant also reports quantifications for multi-gene SEPEPs (C5 SEPEPs), which may provide useful information that is typically missed in the parsimony-based methods. For example, Multiple_SEPEP.2009_C5 was associated with a peptide that could be mapped to two *CDK2* isoforms and one *CDK3* isoform (Fig. 4c). In parsimonious inference, because the two *CDK2* isoforms were also supported by another *CDK2*-specific peptide, this shared peptide was assigned to the *CDK2* isoforms and thus *CDK3* was not quantified. With SEPepQuant, Multiple_SEPEP.2009_C5 was reported to be highly significantly decreased in tumors compared with

NATs (*P* = 4.96e-12). Although this SEPEP was associated with two *CDK2* isoforms and one *CDK3* isoform, it was possible to associate the decrease of this SEPEP specifically to the *CDK3* isoform because another SEPEP (CDK2_SEPEP.3_C3) uniquely mapped to the two *CDK2* isoforms were highly significantly overexpressed in tumors (*P* = 1.00e-15).

Parsimonious inference may also complicate cross-study comparisons. For example, *PYGL* encodes two protein isoforms differing in an exon-skipping event (Fig. 4d). The HCC-TMT study identified many shared peptides and one unique to the long isoform. Accordingly, all peptides were assigned to the long isoform for reporting. In the closely related HCC-label-free study, all identified peptides were shared between the two isoforms. According to the parsimonious principle, all peptides were assigned to the short isoform for reporting. Although significantly decreased expression of *PYGL* in tumors compared with NATs was observed in both studies, and most of the identified peptides were identical in the two studies (Fig. 4d), different protein isoforms reported by the two studies could cause confusions in a cross-study comparison. In addition to reporting one SEPEP associated with the long isoform-specific peptide in the HCC-TMT study, SEPepQuant also reported a non-discriminative SEPEP, which was consistently decreased in tumors compared with NATs in both studies, which helped eliminate potential confusion.

## Protein isoform expression changes during iPSC differentiation into cardiomyocyte

To demonstrate the practical utility of SEPepQuant, we compared SEPepQuant analysis results of the iPSC dataset with those from gene-level quantifications reported by FragPipe. The iPSC dataset was generated by TMT-based shotgun proteomics on iPSC cells cultured over 14 days and harvested daily[21]. As a positive control, we checked the gene and SEPEP-level results of *TPM1*, which was found to have two regulated protein isoforms in the original study. TPM1 showed significant upregulation with culture time based on gene-level quantification by FragPipe (Supplementary Fig. 3a). Consistent with the original study, SEPepQuant identified a group of isoforms recognized by TPM1_SEPEP.2_C3, which differed from the canonical TPM1 sequence at residues 189–212 by mutually exclusive exon (MXE) splicing and were upregulated from day 0 to day 7 and then downregulated from day 7 to day 14 (Supplementary Fig. 3b), as wells as another group of isoforms recognized by TPM1_SEPEP.8_C3, which differed from the canonical *TPM1* sequence at residues 41–80 by MXE splicing and were upregulated at day 14 compared with day 7 (Supplementary Fig. 3c). In addition, SEPepQuant further identified another downregulated isoform group recognized by TPM1_SEPEP.6_C3, which used an alternative translation start site (Supplementary Fig. 3d). Thus, SEPepQuant not only confirmed existing findings but also revealed new information.

Next, we correlated quantifiable genes and SEPEPs with culture time to identify genes and SEPEPs showing monotonic abundance changes during iPSC differentiation into cardiomyocytes. Among the 2028 quantifiable genes with multiple SEPEPs, most showed concordant alterations at gene and SEPEP levels (Fig. 5a and Supplementary Dataset 11). However, 141 SEPEPs from 127 genes showed a significant positive correlation with culture time without matching significance at the gene level, including five SEPEPs from five genes showing a significant negative correlation at the gene level. As an example, PES1_SEPEP.2_C3 was significantly upregulated with culture time, but PES1_SEPEP.1_C4 and the gene-level quantification were significantly downregulated (Fig. 5b). Similarly, 98 SEPEPs from 91 genes showed significant negative correlation with culture time without matching significance at the gene level, including three SEPEPs from three genes showing a significant positive correlation at the gene level. For example, DPYSL3_SEPEP.2_C2, which included a single protein NP_001184223.1 with 5 uniquely identified peptides (Supplementary

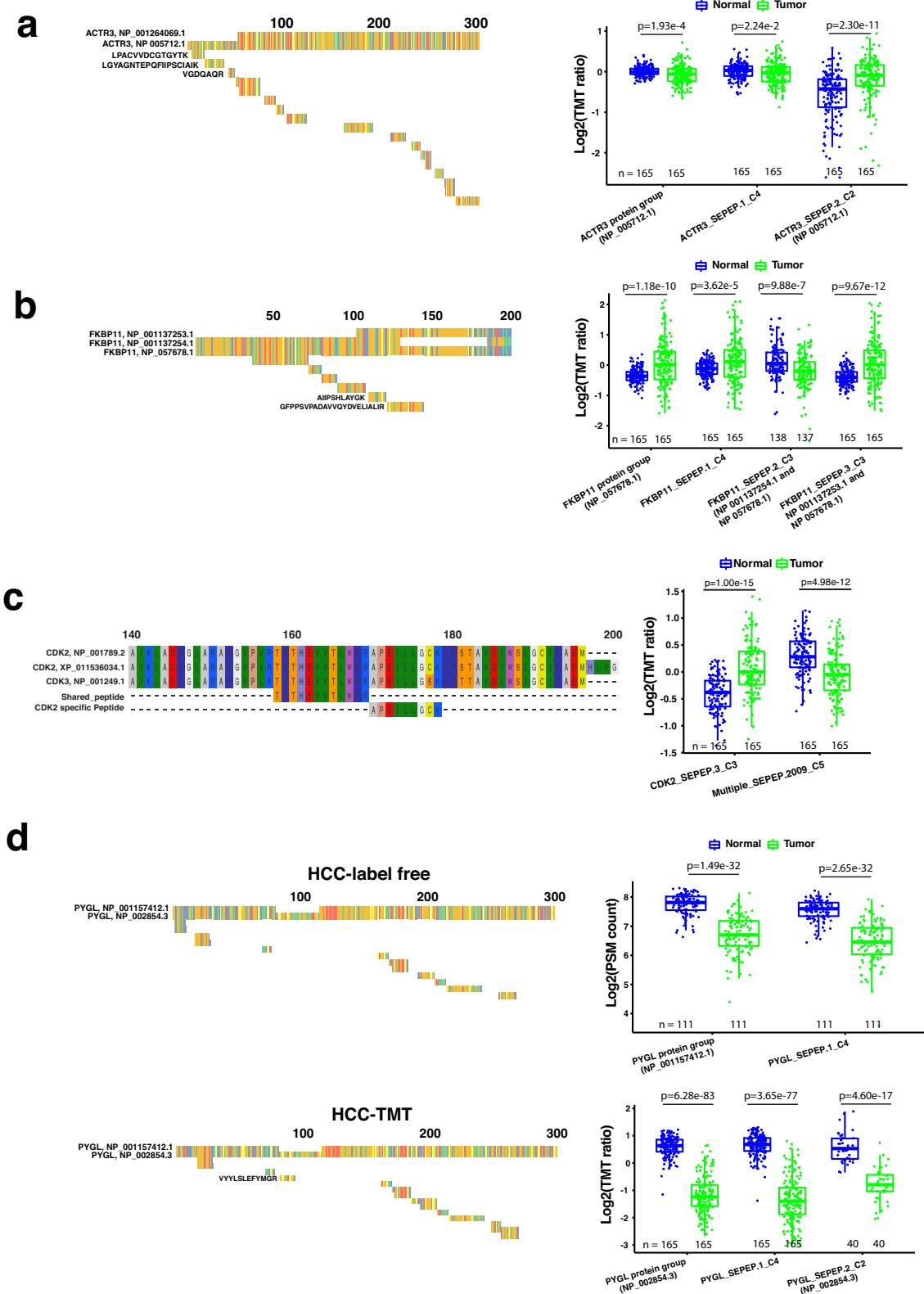

**Fig. 4 | SEPepQuant addresses key limitations of the parsimony-based methods. a** Peptides identified from *ACTR3* within the region spanning the first 300 amino acids of aligned isoform sequences. Tumor and normal comparisons of *ACTR3* SEPEP and protein group levels. **b** Peptides identified from *FKBP11* within the region spanning the first 200 amino acids of aligned isoform squences. Tumor and normal comparisons of *FKBP11* SEPEP and protein group levels. **c** Multi-gene SEPEP provides information on *CDK3*, a gene without uniquely mapped peptides. **d** protein inference of *PYGL* in HCC-label-free and HCC-TMT datasets. The *P* values were computed based on two-sided *t* test. Different colors in protein sequence illustration correspond to different amino acids. For boxplots, centerline indicates the median, box limits indicate upper and lower quartiles, whiskers indicate the 1.5 interquartile range. Source data are provided as a Source Data file.

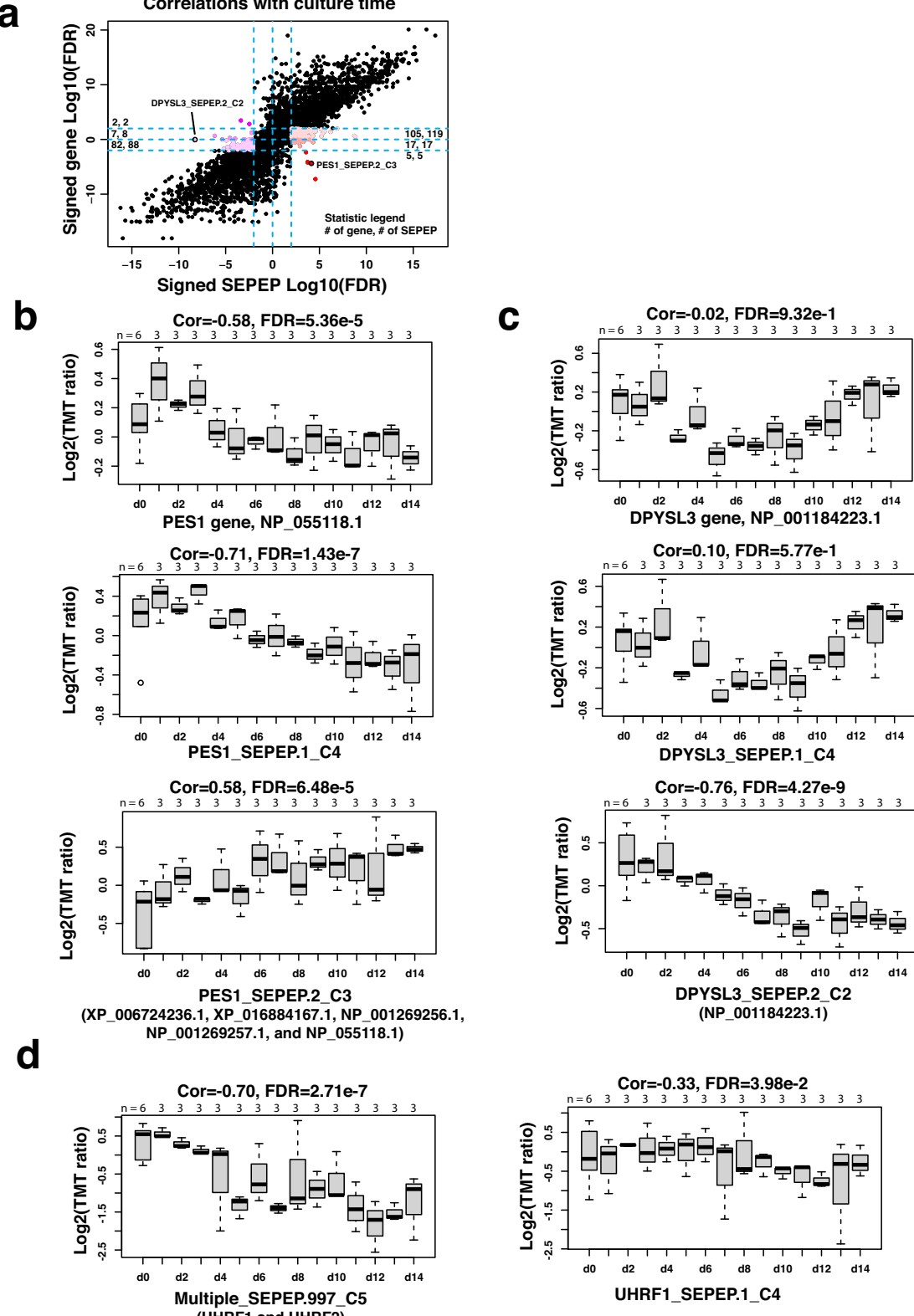

**Fig. 5 | Application of SEPepQuant to an iPSC dataset. a** Scatter plot comparing time-dependent regulations at gene and SEPEP levels. Colored dots are utilized to represent SEPEPs that exhibit a significant correlation with culture time, while lacking a corresponding significance at the host gene level. Additionally, the numbers within each area indicate the count of unique host genes and SEPEPs present in that particular region. **b** Time-dependent regulation of *PES1* at gene and SEPEP levels. **c** Time-dependent regulation of *DPYSL3* at gene and SEPEP levels.

**d** Time-dependent regulation of a multi-gene SEPEP Multiple_SEPEP.997_C5, which is associated with both *UHRF1* and *UHRF2*, and a single-gene SEPEP UHRF1_SEPEP.1_C4, which is specific to *UHRF1*. The *P* values were calculated using Pearson's Correlation, and the Benjamini and Hochberg method was used to adjust *P* values for multiple comparisons. For boxplots, centerline indicates the median, box limits indicate upper and lower quartiles, whiskers indicate the 1.5 interquartile range. Source data are provided as a Source Data file.

Fig. 3e), was significantly downregulated with time; but both DPYSL3_SEPEP.1_C4 and the gene-level quantification showed a pattern of downregulation from day 0 to day 7 and upregulation from day 7 to day 14 (Fig. 5c). This observation was further validated by the RNASeq data obtained from the original publication, despite the limited number of time points profiled (four in total). Specifically, the transcript NM_001197294.2, which corresponds to NP_001184223.1, showed a consistent downward trend from day 0 to day 14, whereas the other transcripts and the gene-level measurements exhibited the lowest values at day 7 but showed an increase by day 14 (Supplementary Fig. 3f).

Our analysis also identified 221 multi-gene SEPEPs showing significant correlation with culture time (Supplementary Fig. 3g). The most significantly positively correlated multi-gene SEPEP, Multiple_SEPEP.122_C5, was associated with four calmodulin-dependent protein kinase genes *CAMK2D*, *CAMK2A*, *CAMK2B*, and *CAMK2G*. The most significantly negatively correlated multi-gene SEPEP Multiple_SEPEP.978_C5 was associated with four zinc finger genes *ZNF93*, *ZNF431*, *ZNF714*, and *ZNF92*. Although it is difficult to attribute the associations to specific genes, these observations still revealed the important roles of these gene families in iPSC differentiation. Moreover, data specific to some genes in a multi-gene SEPEP could be leveraged to improve the interpretation of the association observed for the multi-gene SEPEP. For example, the SEPEP Multiple_SEPEP.997_C5 was associated with both *UHRF1* and *UHRF2*, and it was found to be significantly negatively correlated with time (Fig. 5d). Because UHRF1_SEPEP.1_C1 was not significantly correlated with time (Fig. 5d), it is logical to infer that the significant negative association between Multiple_SEPEP.997_C5 and time was driven primarily by *UHRF2* even though it was not specifically quantified in this dataset.

**Protein isoforms associated with liver cancer development and prognosis**

The HCC-TMT dataset included liver tumor samples and paired NATs from 165 patients, and overall survival information is also available for these patients. We performed tumor vs NAT comparison and survival analysis based on SEPepQuant and gene-level quantifications reported by FragPipe, respectively. In the tumor vs NAT comparison, most of the 4481 genes with multiple SEPEPs showed concordant alterations at gene and SEPEP levels (Fig. 6a and Supplementary Dataset 12). However, 396 SEPEPs from 330 genes showed a significant increase in tumor samples without matching significance at the gene level, including 78 SEPEPs from 69 genes showing a significant decrease in tumor samples at the gene level. Similarly, 392 SEPEPs from 331 genes showed a significant decrease in tumor samples without matching significance at the gene level, including 93 SEPEPs from 85 genes showing a significant increase in tumor samples at the gene level. In the survival analysis, a higher level of concordance was observed between SEPEP-level and gene-level results (Fig. 6b and Supplementary Dataset 13). However, there were still hundreds of SEPEPs showing significant positive or negative associations with survival without matching significance at the gene level. Notably, 18 genes showed significantly increased expression in tumors compared with normal samples as well as significant association with poor prognosis at the SEPEP level, and these associations were not observable at the gene level (Fig. 6c and Supplementary Dataset 14).

One of the 18 was SLK_SEPEP.2_C2, a fully discriminative SEPEP associated with a single protein isoform NP_001291672.1 encoded by the STE20-like serine/threonine-protein kinase gene *SLK* (Fig. 6d–f). Identification and quantification of this SEPEP were based on two junction peptides specific to NP_001291672.1 (Supplementary Fig. 4a). Among the three protein isoforms encoded by *SLK*, SEPepQuant connected this specific isoform to liver cancer development and prognosis, suggesting a critical pro-tumor role of the exon-skipping event.

Although the quantification for the NP_001291672.1-specific SEPEP was very sparse in the independent HCC-label-free dataset, samples with identification of NP_001291672.1-specific peptide KKEEQEFVQK had significantly worse survival compared with samples without identification of this peptide (Supplementary Fig. 4b), which provided independent confirmation of our finding in the TMT dataset. Protein group-level quantification from FragPipe reported two protein groups with representative proteins NP_001291672.1 and NP_055535.2, respectively. However, the protein group represented by NP_001291672.1 showed no significant association with survival in FragPipe quantification (Supplementary Fig. 4c). This is because multiple shared peptides were assigned to this protein group based on the parsimony principle, but shared peptides derived from the other protein group, which had significant association with good prognosis (Supplementary Fig. 4d), could greatly dilute the signal specific to NP_001291672.1.

In order to verify the correlation between the exon-skipping event in *SLK* and tumor initiation and prognosis in HCC, we performed parallel reaction monitoring (PRM) analysis on 20 paired tumor and NAT samples selected from the HCC-TMT study (Supplementary Dataset 14). Among these sample pairs, 10 were obtained from patients who passed away within 12 months of tissue collection (poor prognosis), while the other 10 pairs were obtained from patients who survived for more than 40 months after tissue collection (good prognosis). For the PRM experiment, we specifically chose five peptides that had been previously identified in the TMT study (Supplementary Dataset 14). One of these peptides, KKEEQEFVQK, was found exclusively in the isoform NP_001291672.1, which resulted from exon skipping. Our analysis revealed a significant increase in the abundance of this peptide in tumor samples compared to NATs (Fig. 6g). Moreover, when comparing poor prognosis tumors to good prognosis tumors, we observed a higher abundance of KKEEQEFVQK in the former (Fig. 6h). Conversely, another peptide EVINEVEK, which exclusively mapped to the two exon inclusion isoforms, displayed the opposite pattern. Its abundance was decreased in tumors compared to NATs, and further decreased in poor prognosis tumors compared to good prognosis tumors (Fig. 6g, h). The other three peptides, shared by both the exon-skipping and exon inclusion isoforms, demonstrated either lower levels of increase or even a decrease in abundance when comparing tumor samples to normal samples, as well as when comparing poor prognosis tumors to good prognosis tumors (Fig. 6g, h). These results provide robust evidence confirming the association between the exon-skipping event in SLK and tumor initiation and prognosis in HCC.

Another SEPEP showing significantly increased expression in tumors compared with normal samples as well as significant association with poor prognosis but without concordant changes at the gene level was TF_SEPEP.2_C2, a fully discriminative SEPEP associated with a single protein isoform NP_001054.2 encoded by the transferrin (*TF*) gene (Fig. 6i–k). Identification and quantification of this SEPEP were based on three peptides mapping to the N-terminal region specific to NP_001054.2 (Supplementary Fig. 4e). Among the three protein isoforms encoded by the *TF* gene, SEPepQuant connected this specific isoform to liver cancer development and prognosis. Despite sparse identification of the isoform-specific peptides, the trend was confirmed in the independent HCC-label-free dataset (Supplementary Fig. 4f, g). Protein group-level report from FragPipe only reported one protein group with NP_001054.2 as the representative protein because it can explain all identified peptides (Supplementary Fig. 4e). However, quantification of this protein group based on all peptides was equivalent to gene-level quantification which showed no significant difference in tumor vs normal comparison and no significant association in survival analysis (Fig. 6i, j).

Our analysis also identified 423 and 418 multi-gene SEPEPs with a significant association with good or poor prognosis, respectively

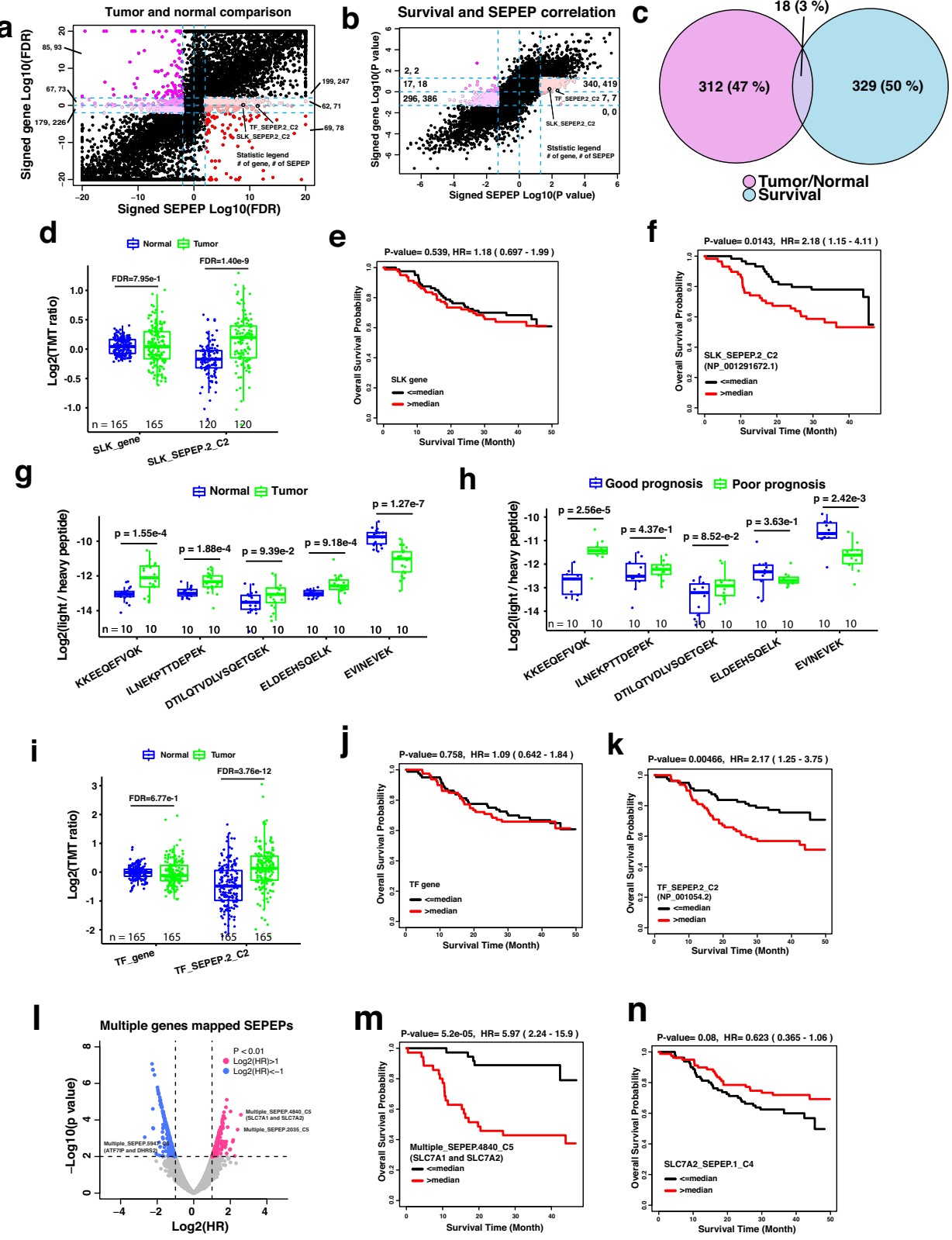

(Fig. 6i). Among them, multiple_SEPEP.5947_C5 showed the lowest hazard ratio. This SEPEP was associated with two genes *ATF7IP* and *DHRS2*, and *DHRS2* has been reported to inhibit cell growth and motility in esophageal squamous cell carcinoma[34]. Multiple_SEPEP.4840_C5 showed the strongest association with poor prognosis. This SEPEP was associated with two genes *SLC7A1* and *SLC7A2* (Fig. 6m). Interestingly, SLC7A2_SEPEP.1_C4 was associated

with slightly longer patient survival (Fig. 6n), suggesting that the association between Multiple_SEPEP.4840_C5 and poor prognosis was driven by SLC7A1. Consistent with this inference, Multiple_SEPEP.4840_C5 showed significantly higher abundance in tumors compared with normal samples, but SLC7A2_SEPEP.1_C4 was significantly decreased in tumors (Supplementary Fig. 4h). Despite the lack of detection of gene-specific peptides for *SLC7A1*, SEPepQuant

**Fig. 6 | Application of SEPepQuant to an HCC-TMT dataset. a** Scatter plot of tumor versus normal comparison results at gene and SEPEP levels. **b** Scatter plot of survival association results at gene and SEPEP levels. Colored dots in (**a**, **b**) are utilized to represent SEPEPs that exhibit a significant test result, while lacking a corresponding significance at the host gene level. In addition, the numbers within each area indicate the count of unique host genes and SEPEPs present in that particular region. **c** Overlap of genes showing significantly increased expression in tumors compared with normal samples as well as significant association with poor prognosis at the SEPEP level but not at the gene level. **d** Tumor versus normal comparisons based on *SLK* gene and SLK_SEPEP.2_C2 abundance, respectively. **e**, **f** Kaplan–Meier plots comparing overall survival for patients stratified by the median *SLK* gene-level abundance and median SLK_SEPEP.2_C2 abundance, respectively. **g** PRM abundance comparison of selected peptides between tumor and matched normal samples. **h** PRM abundance comparison of selected peptides between selected patients with poor and good prognosis. **i** Tumor versus normal comparisons based on TF gene and TF_SEPEP.2_C2 abundance, respectively. **j**, **k** Kaplan–Meier plots comparing overall survival for patients stratified by the median TF gene-level abundance and median TF_SEPEP.2_C2 abundance, respectively. **l** Associations between survival and multi-genes SEPEPs. **m**, **n** Kaplan–Meier plots comparing overall survival for patients stratified by the median Multiple_SEPEP.4840_C5 abundance and median SLC7A2_SEPEP.1_C4 abundance, respectively. For boxplots, *P* values were calculated using two-sided Student's *t* test, the Benjamini and Hochberg method was used to adjust *P* values for multiple comparisons, centerline indicates the median, box limits indicate upper and lower quartiles, whiskers indicate the 1.5 interquartile range. For survival analysis, *P* values were calculated using Kaplan–Meier test. Source data are provided as a Source Data file. HR hazard ratio.

results clearly suggested a pro-tumor role of this gene, which was previously reported in ovarian cancer[35].

## Discussion

Shotgun proteomics has become an essential tool for protein identification and quantification in biomedical research. However, shotgun proteomics-based protein isoform identification and quantification remains an open challenge, hampering a thorough understanding of protein isoform regulation and their roles in normal and disease biology. We developed SEPepQuant, a graph theory-based approach that uses groups of structurally equivalent peptides in a peptide-protein-gene tripartite graph, instead of protein groups or gene groups, as the identification and quantification unit to enable comprehensive protein isoform characterization in shotgun proteomics. In three experimental datasets, SEPepQuant identified 5.8–33.8 times more genes with multiple quantification units compared with that by parsimony-based protein inference (Fig. 3). For genes with multiple SEPEPs, 35.1–79.8% had at least one SEPEP with a below 0.5 correlation with the corresponding gene abundance, suggesting extensive isoform-specific regulation. Indeed, analysis based on SEPepQuant quantification results revealed more than 100 genes with protein isoform-level regulation during cardiomyocyte differentiation and hundreds of protein isoform-level regulatory events with significant associations to liver cancer development and prognosis.

Parsimony-based protein inference was introduced at the early stage of proteomics research to address the problem of overreporting the number of identified proteins in shotgun proteomics studies[16,17], and it has since become the dominant method in the field. However, the consequence of this method on protein quantification has not been formally evaluated. Our analysis in this paper revealed several key limitations associated with the parsimony-based methods, including ignoring protein isoforms and genes without uniquely identified peptides, incorrect or inaccurate quantification of protein isoforms by simply assigning shared peptides to isoforms with the largest number of identified peptides, and complicating cross-study comparisons because of different reporting isoform selection driven by minor changes in peptide detection in different studies. We showed that SEPepQuant is able to address these limitations, leading to more comprehensive and accurate analysis of protein isoforms.

To reduce the number of degenerate peptides, a commonly employed strategy is the utilization of the UniProt canonical database. However, due to its limited scope of ~20,000 proteins, this database only encompasses a single canonical sequence for most genes, rendering it unsuitable for investigating protein isoform regulation. In our study, we sought to facilitate a comprehensive exploration of protein isoforms by employing the RefSeq database that encompasses both curated proteins (NP and YP) and predicted proteins (XP), resulting in a total of 140,000 entries. When a SEPEP includes both curated and predicted proteins, it makes sense to focus on the curated proteins in further investigation. However, when a SEPEP includes only predicted proteins, it provides direct experimental evidence for the predictions. Opting for a more conserved protein database, such as one exclusively composed of curated proteins, may reclassify certain higher-class SEPEPs into lower classes. However, this approach could potentially overlook regulatory mechanisms involving predicted isoforms. Alternatively, when matched RNASeq data is available, the utilization of customized protein databases derived from such data represents the optimal choice. Notably, SEPepQuant can also be utilized alongside customized databases.

SEPepQuant identified hundreds of protein isoform-level regulatory events from both the iPSC and liver cancer datasets, highlighting widespread impact of protein isoform-level regulation in normal and disease processes. Notably, 18 genes showed significantly increased expression in liver tumors compared with normal samples as well as significant association with poor prognosis at the SEPEP level but not at the gene level. Among these, *SLK* encodes a kinase that promotes apoptosis[36]. The pro-tumor SEPEP of *SLK* is a fully discriminative SEPEP and thus the pro-tumor effect could be attributed to the associated protein isoform NP_001291672.1. Compared to the longer *SLK* isoform NP_055535.2, this short isoform has a skipped exon, which encodes a section of a coiled-coil domain that mediates homodimerization to enhance *SLK* activity[36,37]. Therefore, this exon skipping may lead to reduced *SLK* activity and decreased apoptosis to facilitate tumor progression. Consistently, a recent analysis of RNASeq data from melanoma tumors has shown that expression of the long isoform is decreased, whereas the short isoform is increased in metastatic tumors compared with primary tumors, suggesting a role of the exon skipping in facilitating metastasis[38]. Transferrin is another gene found to be correlated with poor prognosis at the SEPEP level but not gene level. This pro-tumor SEPEP is also a fully discriminative SEPEP associated with NP_001054.2, the longest isoform of the transferrin gene. Transferrin is synthesized primarily in the liver and secreted into serum, with a half-life of eight days in the serum[39]. The unique N-terminal sequence of this isoform and the long half-life of the protein make it a promising candidate for serum biomarkers of liver cancer prognosis for further investigation.

In summary, our analysis provides strong evidence to support a critical and widespread role of protein isoform regulation in normal and disease processes, and SEPepQuant is expected to have broad applications to biological and translational research to boost scientific discoveries.

## Methods

### Ethical statement

Protein lysates used in the PRM experiments were previously prepared[29] from liver cancer specimens collected from patients who underwent surgical resection at Zhongshan Hospital, Fudan University. The study was approved by the Research Ethics Committee of Zhongshan Hospital (B2017-060), and written informed consent was obtained from each patient.

## Protein database

RefSeq gene annotation and matched protein database with all NP, XP, and YP protein sequences were selected for this study (downloaded on 06/28/2020)[39,40].

## In silico digestion

Protein sequences were cut after any K and R except followed by P using in-house script. Up to two missed cleavage sites were allowed in the in silico digestion[41,42]. The resulting peptide sequences with length <7 bp or >50 bp were excluded from our analysis.

## Protein sequence similarity

Protein sequences for genes with multiple protein isoforms were aligned using clustalw2 with default parameters[43]. Then, clustalo was applied to above multiple sequence alignments to calculate sequence similarity with percent identities as sequence distance[44]. Pairwise sequence similarities were extracted from the clustalo percentage identity matrix.

## Experimental datasets

Three published datasets were selected to cover different sample types and proteomics platforms. The iPSC dataset is a human cell line dataset generated on a 10-plex TMT platform. In this study, the human iPSC cells were cultured over 14 days and harvested daily for proteomics analyzing[28]. The HCC-TMT dataset includes 165 tumor samples from hepatocellular carcinoma patients and matched adjacent normal tissues, and the data were generated on a 11-plex TMT platform[29]. The HCC-label-free dataset includes 111 tumor samples from hepatocellular carcinoma patients and matched adjacent normal tissues, and the data were generated on a label-free platform[29,30].

## Database searching and protein quantification using FragPipe

The comprehensive shotgun proteomics analysis pipeline FragPipe (https://fragpipe.nesvilab.org/; V17.0) with MSFragger (v3.4) and philosopher (v4.2.1) was used for database searching and protein group-level and gene-level quantification for all three datasets[14,15]. MaxQuant (v2.3.1.0) was only applied to the iPSC dataset[11]. The same parameters and modifications from the original publication were used for database searching. PeptideProphet and ProteinProphet with default parameters were used for peptide validation and protein inference[45,46]. PSM count was used for label-free data quantification, and TMT ratio was used for quantification in TMT-based datasets. Median centering was used as the normalization method for TMT-based datasets.

## SEPEP quantification and normalization

For label-free data, the sum of PSMs of all peptides from a SEPEP was used to quantify the SEPEP. For TMT-based data, the median value of the TMT ratios of all peptides from a SEPEP was used to quantify the SEPEP. Normalization factors derived from median centering of the gene-level analysis described above were used to normalize SEPEP quantifications.

## PRM data generation and quantification

The tissues were from a published study[29] and were lysed in SDS lysis buffer (4% SDS, 100 mM Tris-HCl, 0.1 M DTT, pH 7.6) previously. Filter-aided sample preparation (FASP) procedure was used for peptide preparation with all centrifugation steps at $12,000 \times g$ at 25 °C. Briefly, a total of 50 µg proteins for each sample were loaded in 10-kDa centrifugal filter tubes (Millipore), washed twice with 200 µL UA buffer (8 M urea in 0.1 M Tris-HCl, pH 8.5), alkylated with 50 mM iodoacetamide in 200 µL UA buffer for 30 min in the darkness, washed thrice with 200 µL UA buffer again and finally washed thrice with 200 µL 50 mM NH$_4$HCO$_3$. Proteins were digested at 37 °C for 18 hr with trypsin (Promega) at a concentration of 1:50 (w/w) in 50 mM NH$_4$HCO$_3$. After

digestion, peptides were eluted by centrifugation and acidated by trifluoroacetic acid (TFA) with a final concentration of 1%. Then the peptides were purified using C18 stage-tips, and the elutes were subjected to vacuum centrifugation dryness. The peptides were resolved in 20 µL 0.1% formic acid (FA) and the concentration was determined using NanoDrop.

Six standard peptides were synthesized with heavy-isotope labeled lysine in the C-terminal in GL Biochem (Shanghai) Ltd. The synthetic peptides were first resolved in 20% ACN and diluted using 0.1% FA for further use. For the PRM experiment, 1 µg peptides of each sample were mixed with the six standard peptides (1 pmol for each peptide). The peptides were separated using a home-made micro-tip C18 column (75 µm × 200 mm) packed with ReproSil-Pur C18-AQ, 3.0-µm resin (Dr. Maisch GmbH, Germany) on a nanoflow HPLC Easy-nLC 1200 system (Thermo Fisher Scientific), using a 60 min LC gradient at 300 nL/min. Buffer A consisted of 0.1% (v/v) FA in H$_2$O and Buffer B consisted of 0.1% (v/v) FA in 80% acetonitrile. The gradient was set as follows: 1–32% B in 43 min; 32–45% B in 7 min; 45–100% B in 2 min; 100% B in 8 min. The PRM analyses were performed on a Q Exactive HF-X mass spectrometer (Thermo Fisher Scientific) using a scheduled mode to monitor both the "light" and "heavy" parent ions of the six peptides. The retention time of each peptide was determined in a preliminary experiment and the ±3 min of the retention time for each peptide was targeted for analysis. Information of the five peptides is shown in Supplementary Dataset 14.

The MS parameters were set as follows: The spray voltage was 1800 V in positive ion mode, and the ion transfer tube temperature was set at 320 °C. MS data acquisition was performed using Xcalibur software in profile spectrum data type. The MS1 full scan was set at a resolution of 120,000 @ $m/z$ 200, AGC target 3e6, and maximum IT 50 ms by orbitrap mass analyzer (350–1500 $m/z$). The MS/MS scans of the target peptide were generated by HCD fragmentation at a resolution of 15,000 @ $m/z$ 200, AGC target 1e5 and maximum IT 100 ms. Isolation window was set at 1.0 $m/z$. The normalized collision energy (NCE) was set at NCE 27%, and the loop count was set at 12.

The MS data were processed using the Skyline software. The chromatographic peak of each peptide was manually checked. For each peptide, five product ions with the top intensities were selected for quantification analysis. The ratios of light to heavy peptides were reported and used to perform the quantification comparison among samples.

## Statistics and reproducibility

This study did not employ a statistical method for predefining sample size, because all cohorts used in our investigation were derived from previously published studies. To validate our findings based on global proteomics data, we performed targeted PRM analysis on 20 liver cancer tissues. These tissues were selected based on their high or low NP_001291672.1 abundance, and protein lysate availability from the original publication. The primary focus of this study revolves around the identification and quantification of protein isoforms using datasets available in existing literature. As such, no data was excluded from the analyses, and randomization was not applicable. The Investigators were not blinded to allocation during experiments and outcome assessment.

## Reporting summary

Further information on research design is available in the Nature Portfolio Reporting Summary linked to this article.

## Data availability

The human iPSC[28] and HCC-label-free[30] datasets were downloaded from PRIDE (www.ebi.ac.uk/pride) with accession numbers PXD013426 and PXD006512. The PRM proteomics data of 20 selected samples generated in this study have been deposited in the PRIDE

database under accession code PXD044809. The HCC-TMT[29] proteome data was downloaded from NODE (https://www.biosino.org/node) by accession number OEP000321. ProteoWizard (vc143) was used to transfer raw files to mzML files[47]. Processed quantification data and downstream analysis results were included in Supplementary Datasets 1–14. Source data are provided with this paper.

## Code availability

SEPepQuant is open source under an Apache 2.0 license and can be found at https://github.com/bzhanglab/SEPepQuant. The version used in the manuscript has been publicly released (https://github.com/bzhanglab/SEPepQuant/releases/tag/v1.0.0-alpha). The source code has also been placed on the Zenodo platform [https://doi.org/10.5281/zenodo.8258298][48].

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

## Acknowledgements

This study was supported by grants U24 CA210954, U24 CA271076, and R01 CA245903 from the National Cancer Institute (NCI), the Cancer Prevention & Research Institutes of Texas (CPRIT) award RR160027, and funding from the McNair Medical Institute at The Robert and Janice McNair Foundation to B.Z. The work on PRM validation was supported by grants from Shanghai Young Excellent Academic Leader Program (20XD1424900) and the Shanghai Municipal Science and Technology Major Project to H.Z. B.Z. is a CPRIT Scholar in Cancer Research and a McNair Scholar.

## Author contributions

Y.D. and B.Z. conceived the project and designed the study. Y.D. processed proteomics data and performed the analyses with help from B.Z., X.Y., and L.K.O. Y.L., H.W.Z., H.Z., and Q.G. performed experimental validation in the study. Y.D. and B.Z. wrote the manuscript. All authors edited and approved the manuscript.

## Competing interests

B.Z. received consulting fees from AstraZeneca. The remaining authors declare no competing interests.
