## [Peer Review File · Nature Communications]

SEPEPQuant enhances the detection of possible isoform regulations in shotgun proteomicsReviewer #1 (Remarks to the Author):

This study concerns the protein inference problem in shotgun proteomics as applied to protein isoform identification. Because many isoforms share similar sequence, it is not always clear which protein/isoform emits a particular observed peptide. Here the authors use a graphical structure to categorize peptides based on whether they are unique to a protein isoform, common to all isoforms, or are grouped to subsets of isoforms. etc. In reanalyzing existing data sets they show that some additional information can be gained with this approach over the more common unique protein or razor protein strategies. An important advance here is that more SEPEP can be found that show significant changes in development and disease so can recover more information from proteomics data set. Understandably this also comes with some drawbacks including the increase multiple testing burden, and also some of the results may be less actionable since it is not known which actual protein isoform is responsible for the changes.

Overall I think this manuscript addresses an important area, although it could also use some revision. The way the manuscript is currently written doesn't seem to fully describe what the strategy tries to do. The overall approach here seems to be a mix of protein inference and using peptide-level data of unresolved protein groups to discover potential regulation which would be similar to what has been attempted in PQQ (Forshed et al. Mol Cell Proteomics 2011) and PeCorA (Dermitt et al. J Proteome Res 2020). Part of the results (e.g., Figure 5d) seems to suggest we should quantify first and forego protein inference altogether. As the authors will know previous protein inference works have also used graphical models (e.g., the bipartite graphs in SCAMPI - Gerster et al. Mol Cell Proteomics 2014) for protein quantification, and gpGrouper arguably took this further by distributing protein intensities across unique groups. Implicitly these methods also address protein isoforms because they concern peptides that are shared by one or more proteins. So there should be a clearer comparison and contrast with prior methods and the manuscript should go into greater depth on which part of the strategy is specific to protein isoforms.

1. Page4, the authors state "Our method is fundamentally different from existing approaches because we use peptide groups determined from graph modeling, rather than protein groups or gene groups, as the quantification unit." This sentence is somewhat difficult to parse. If one uses, say, Percolator, one will get a list of protein accessions from which each identified peptide appears, so these protein groups are essentially peptide groups, which can be used for label free or TMT quantification. Is the major novelty here a categorization scheme to show whether those accessions in a protein group come from the same gene, or the modification of the picked protein method to include shared protein groups and not just unique groups? Ideally this should be more clearly stated.

2. Page 5, the authors used an in silico analysis (digesting RefSeq protein entries virtually) to estimate the proportion of peptides that may map to one or more isoforms, but how this analysis is done is different from real database search parameters which makes it hard to interpret the results. Only fully tryptic peptides with no miscleavages are used, whereas in a real world scenario the search parameter would include semi-tryptic ends and 1-2 miscleavages allowed which depending on the experiment could account for a substantial portion of peptides. This is especially important for isoform analysis since, as this same group has previously very nicely shown, lysines and arginines are enriched in exon boundaries so often an isoform is discerned by a miscleaved cross-junction peptide. Likewise, peptides of 7 and 8 amino acids should either be removed from consideration or interpreted with care to be in line with community guidelines calling for peptides with at least 9 aa to provide strong evidence of non-canonical sequences. In my opinion this section should either be expanded to match realistic experimental conditions or removed.

3. Page 6, the authors wrote that "Percentages of the single isoform peptides in these studies were slightly higher than that in in silico digestion, whereas an opposite trend was found for multi-isoform peptides (Fig. 1d), suggesting competition among multiple isoforms may reduce transcriptional and/or translational efficiency." This statement is somewhat confusing, because the in silico digestion treats all proteins as equal, whereas in the experimental data there is a dynamic range of concentration between different proteins. It could just be that proteins with fewer isoforms also tend to have higher natural abundance so would be detected more often (e.g., see the two miscleavage peptide KKEEQEFVWK for NP_001291672.1)

4. More generally, it would seem that both the in silico analysis and the overall inference method are highly dependent on which database is being supplied. Which peptide is partially discriminant or fully isoform discriminant will change based on how many isoforms are in the database. If the

database is over-annotated and contains many redundant/unverified isoforms that don't exist in a particular sample then it might stack the odds against unique peptides or razor peptides methods. The RefSeq database being used including NP and XP entries is quite big (~140k sequences) so one would expect there are more class 3 and class 4 SEPEPs than if one were to use UniProt or a custom database. How that would affect the analysis and results should probably be investigated more thoroughly.

5. The title of the manuscript is somewhat ambiguous. It states "comprehensive protein isoform characterization", but of course the isoforms are not necessarily resolved in the method so they are not characterized per se, e.g., a class 3 SEPEP may show some differential regulation but it is not clear which of the actual protein isoform is responsible. A more precise description along the line of recovering potential isoform regulations should be considered.

6. Page 16: "For genes with multiple SEPEPs, 35.1% - 79.8% had at least one SEPEP with a below 0.5 correlation with the corresponding gene abundance, suggesting extensive isoform-specific regulation." This is an important observation, but should be placed in fuller context including background variance. What is the range in correlations between individual peptides in the non-discriminative SEPEPs, and is this a function of protein abundance and peptide intensity in the mass spec? To me this would be really important especially for inferring isoforms without any unique peptide whatsoever, e.g., in the example of ACTR3 on page 9, where the short isoform is completely subsumed in the long isoform. Could the differential trends between peptides be more simply explained by post-translational modifications or variance due to low peptide abundance? Is there corroborating evidence at the RNA level?

Reviewer #2 (Remarks to the Author):

In this manuscript, the authors present SEPEPQuant for SEPEP-level quantification and identification of differentially expressed SEPEPs using bottom-up MS. SEPEPQuant can provide additional information for proteoform quantification, and SEPEPQuant analyses of several data sets demonstrate that it identified proteoform-level regulation events that were missed by protein-level analysis.

Major comments.

1. Several proteogenomic methods have been proposed to analyze splice junction peptides, which are related to SEPEP and should be cited in Section Introduction. For example,

Woo et al, Proteogenomic Analysis Reveals Multiple Peptide Mutations and Complex Immunoglobulin Peptides in Colon Cancer, *Journal of Proteome Research*, 215, 3555-3567.

2. Parsimony-based protein inference can be easily extended to parsimony-based proteoform inference. The difference between SEPEP and parsimony-based proteoform inference should be clarified and discussed. In Fig 3d, the SEPEP approach needs to be compared with parsimony-based protein inference and proteoform inference.

3. In SEPEP FDR control, the sentence "Specifically, a SEPEP is considered a target hit if the highest scoring peptide in the SEPEP is from a forward protein sequence." is confusing. All peptides in a SEPEP are structurally equivalent. If the highest-scoring peptide is from target proteoforms, all other peptides in the same SEPEP should be from the same target proteoforms. Is it possible that a peptide is shared by target and decoy proteoforms. In this case, how to determine if the SEPEP is a target or decoy identification.

4. In TMT data, the peptides of a SEPEP identified from one sample may be different from those of the same SEPEP identified from another sample. When tens of samples are analyzed, the problem becomes more complicated. It is unclear how to do SEPEP level quantification for TMT data.

5. Fig 4 shows several examples of differentially expressed SEPEPs. The iPSC and HCC data sets contain both RNA-Seq and MS data. Can the SEPEP abundances of the isoforms in Fig. 4 be compared with their transcript expression levels?

Minor comments

1. In Fig. 2b, protein information in the tripartite graph is not needed to identify SEPEPs, which should be pointed out in the manuscript.
2. The word "group" is not included in SEPEP, which makes it unclear if a SEPEP is a peptide or a peptide group. The authors may consider changing SEPEP to include the word "group."
3. For Fig 3e, the authors gave an explanation: "more stringent FDR control caused by the much larger number of candidate SEPEPs compared with candidate genes." The explanation might be incorrect. The reason might be that more stringent FDR control was caused by the less number of candidate SEPEPs compared with the candidate peptides.
4. The resolution of Fig. 5a, Fig. 6a, Fig.6b is low.

Reviewer #3 (Remarks to the Author):

The article 'SEPEPQuant enables comprehensive protein isoform characterization in shotgun proteomics' by Dou et al. introduces a new methodology to characterize isoforms from shotgun proteomics data. The method addresses a very important topic, as isoforms of many proteins have been shown to regulate biological processes differently – sometimes in opposing manners – yet many proteomic studies ignore isoforms due to a lack of suitable methods to distinguish isoforms.

The authors propose a new graph-based method, where they represent the relations between genes, proteins and peptides through a tripartite graph. This relatively simple representation allows the authors to identify isoforms of proteins not possible with a parsimonious method to which the authors compare their approach. The paper has a big potential to advance the field and help experimental researchers to identify protein isoforms, which may have a big impact on the investigated biological system. The biggest limitation, in my opinion, is the current lack of independent verification that their method works. For example, an independent experiment could be performed to confirm the presence and quantity of the identified isoforms as predicted by the author's method. For example, Western Blots of isoforms identified could be performed to confirm that the methods the authors propose give compatible results with the Western Blots (if suitable antibodies are available for the proteins discussed in the manuscript. However, the method was applied to big datasets so surely some of those should allow for independent measurements with Western Blot or otherwise). Besides this major comment, I include several smaller comments below that should be addressed before the manuscript can be considered for publication.

The authors compare their approach to 'the parsimonious' approach. But as they say, there are different implementations by different groups. How much difference do those published methods have in terms of isoform characterisation? Why did the authors only pick FragPipe as a comparison, and not other tools? A more systematic comparison of the different tools available for at least some of the data presented would strengthen the present paper.

Abstract: protein isoform characterization is inhibited: inhibited sounds a bit strong. Maybe: ... is challenging due to the extensive ...

p 3: the extent to which transcript isoform complexity propagates to the proteome remains controversial: This sentence seems to say that not all transcript isoform complexity may translate to protein complexity, i.e. complexity is reduced. This can happen, but there can also be higher complexity on the protein level due to PTM etc.

p6 This could be explained by possibly higher abundance of these

peptides because they can be derived from multiple genes: I do not fully understand this explanation. If a peptide is derived from multiple genes, why is this not reflected in the in silico study? The RefSeq protein database should then also include these peptides multiple times. I do not understand why this explains a mismatch between in silico experiments and real data.

Fig 1 d, e: the caption should briefly mention what we see. In general, most captions in this paper could be a bit more detailed

P6, middle: in consistent: remove: in

P8 middle: the 50% threshold for missing data: how robust is the method when that threshold is changed? Also is the same threshold applied to the parsimonious approach?

Fig 3f: state exactly over what data the correlations are calculated (in the text or supplement)

Fig 4: the choice of investigated proteins should be motivated, as otherwise, one could get the impression that the authors cherry-picked proteins with a big difference in their method to the parsimonious approach. Tables showcasing all proteins should also be presented in the supplement

Fig 4a: I do not fully understand if the statement in the text 'whereas the parsimonious inference provided only incorrect information for the long isoform.' is trivial? Is this solely based on the fact that the parsimonious approach assigns every peptide to the long isoform, so naturally that information is incorrect as it also includes all peptides from the short isoform? If this is the case, I would not say that this approach shows anything incorrectly about the long isoform. This approach does not show the long isoform at all! It shows data about a group of proteins, one cannot really say whether this is the long or short isoform.

Fig 5/Supp Fig 2/ p12: The investigation of correlations of different isoforms with time is very promising. However, despite statements such as 'these observations still revealed important roles of these gene families in iPSC differentiation', the section contains few ideas of what we could learn about the biology of stem cell differentiation from these findings. While this is not the major objective of the paper, I believe the method could gain more acceptance if some striking biological interpretations could be obtained. More importantly, independent experiments should confirm the new findings

5a: the authors identified gene/SEPEP pairs that are not correlated in tissue time (and similar analysis for tumour versus non-tumour in 6a). Did the authors perform some correction for multiple testing to rule out that with so many genes investigated, it did not occur just by chance that some parts are anticorrelated?

Re: NCOMMS-22-41664 “SEPEPQuant enables comprehensive protein isoform characterization in shotgun proteomics”

REVISIONS IN RESPONSE TO REVIEWERS' COMMENTS

We thank the reviewers for the insightful comments and constructive suggestions. We have considered all comments and suggestions and revised the manuscript accordingly. Please see below for a point by point response to each of the points made by the reviewers. For your convenience, we have highlighted the major changes in green in the revised manuscript. Reviewers' comments, our response, and new text in the revised manuscript are colored in black, blue, and green, respectively, in this document.

REVIEWER COMMENTS

Reviewer #1 (Remarks to the Author):

This study concerns the protein inference problem in shotgun proteomics as applied to protein isoform identification. Because many isoforms share similar sequence, it is not always clear which protein/isoform emits a particular observed peptide. Here the authors use a graphical structure to categorize peptides based on whether they are unique to a protein isoform, common to all isoforms, or are grouped to subsets of isoforms. etc. In reanalyzing existing data sets they show that some additional information can be gained with this approach over the more common unique protein or razor protein strategies. An important advance here is that more SEPEP can be found that show significant changes in development and disease so can recover more information from proteomics data set. Understandably this also comes with some drawbacks including the increase multiple testing burden, and also some of the results may be less actionable since it is not known which actual protein isoform is responsible for the changes. Overall I think this manuscript addresses an important area, although it could also use some revision.

Response: We thank the reviewer for recognizing the significance of our manuscript and providing constructive revision suggestions.

The way the manuscript is currently written doesn't seem to fully describe what the strategy tries to do. The overall approach here seems to be a mix of protein inference and using peptide-level data of unresolved protein groups to discover potential regulation which would be similar to what has been attempted in PQQ (Forshed et al. Mol Cell Proteomics 2011) and PeCorA (Dermitt et al. J Proteome Res 2020). Part of the results (e.g., Figure 5d) seems to suggest we should quantify first and forego protein inference altogether. As the authors will know previous protein inference works have also used graphical models (e.g., the bipartite graphs in SCAMPI - Gerster et al. Mol Cell Proteomics 2014) for protein quantification, and gpGrouper arguably took this further by distributing protein intensities across unique groups. Implicitly these methods also

address protein isoforms because they concern peptides that are shared by one or more proteins. So there should be a clearer comparison and contrast with prior methods and the manuscript should go into greater depth on which part of the strategy is specific to protein isoforms.

Response: We thank the reviewer for this important comment. We updated the Introduction section to provide background information on methods mentioned in this comment as well as other related methods. We also highlighted the unique innovation of our method, which is the introduction of SEPEP as the quantification unit.

New Text, Pages 4 and 5:

“To address the challenge in protein isoform discrimination, several methods have been developed. One solution is to perform gene-based quantification, which is implemented in tools such as gpGrouper¹⁹ and FragPipe and used in some recent studies^{20–22}. gpGrouper uses quantities of gene-specific peptides to guide the split of quantities of shared peptides. Because only a small proportion of peptides are shared between genes, it makes full use of both unique and shared peptides to produce gene level quantification with demonstrated accuracy¹⁹. However, protein isoform information is ignored in this approach. Along the same line, SCAMPI uses statistical modeling to generate quantification for individual proteins using both unique and shared peptides²³. This approach was demonstrated when proteomics data were searched against the UniProt canonical database in which only canonical protein sequences are included, which means a single canonical sequence for most genes. For example, although there are eight annotated isoforms of TP53 in UniProtKB, the canonical database only included a single canonical isoform. Therefore, isoform reduction occurred in database construction, and the challenge addressed was primarily to distribute quantities of peptides shared by different genes. The method's ability to handle search results from a comprehensive protein database is uncertain because the inclusion of canonical and alternative isoform sequences leads to a substantial reduction in the number of isoform-specific peptides, which are crucial for accurately distributing the quantities of peptides shared between isoforms. Moreover, it is also unclear whether the method can be directly applied to ratio data generated from labeled experiments such as tandem mass tag (TMT)-base experiments.

Methods have also been developed based on the assumption that the quantitative pattern of peptides derived from one protein will correlate over several samples. Protein Quantification and Peptide Quality Control (PQPQ) selects peptides correlating over samples to improve the quantitative accuracy and precision²⁴, whereas Peptide Correlation Analysis (PeCorA) focuses on outlier peptides to reveal differential proteoform regulation²⁵. These methods require multiple samples for analysis, and peptides annotated to more than one protein are excluded from the analysis. Therefore, many peptides would be excluded when a comprehensive protein database covering both canonical and alternative isoform sequences are used for database searching.

Leveraging RNASeq data from matched samples, the CustomProDB approach constructs a customized protein database with reduced number of protein isoforms by excluding isoforms with low transcript abundance in matched RNASeq data²⁶. This method also enables novel protein isoform identification, but the quantification challenge remains because each gene may still have

multiple RNASeq data-supported isoforms. Based on a strong assumption that each gene only has a dominant isoform, Liu et al.¹⁰ used matched RNASeq data to select one isoform with the highest transcript abundance for each gene. Woo et al.²⁷ and Lau et al.²⁸ focus on novel protein isoforms by identifying and quantifying peptides mapped to novel isoform junctions detected based on RNASeq data. While these methods are appealing, their utility in the majority of proteomic studies is limited by the prerequisite of matched RNASeq data.”

New Text, Page 5-6:

“To tackle the challenge of protein isoform characterization and leverage the potential opportunities, we extend the bipartite graph representation of peptide-protein relationships¹⁷ to a tripartite graph for a comprehensive representation of the peptide, protein, and gene relationships. From the tripartite graph, we define a new quantification unit called Structurally Equivalent PEptides (SEPEPs). These SEPEPs consist of peptide vertices that are connected to precisely the same set of protein vertices within the graph and are thus structurally equivalent in the graph. To facilitate downstream interpretation, we further divide the SEPEPs into five classes based on their patterns of connections to source proteins and genes in the tripartite graph. The introduction of SEPEPs as the quantification unit represents a significant innovation. It fundamentally differs from existing quantification approaches that employ parsimonious protein groups, individual genes, individual proteins, or correlated peptides from individual proteins, as the unit of quantification. While using peptides mapping exclusively to a single protein for quantifying that specific protein, as implemented in PQQ and PeCorA, provides accurate quantification, it excludes many peptides that are shared by multiple protein isoforms. On the other hand, when parsimonious protein groups or genes are employed as the quantification units, the isoform-specific information available from shotgun proteomics data is often suppressed or lost. By shifting the quantification unit to peptides that exclusively map to a group of protein isoforms that are indistinguishable based on the identified peptides, our SEPEP-based method can leverage all confidently identified peptides, including those mapping to multiple proteins or even multiple genes. Moreover, this approach retains and utilizes all the available isoform-distinguishable information present in the data, thus enhancing the detection of possible isoform regulations in our analysis.”

1. Page4, the authors state "Our method is fundamentally different from existing approaches because we use peptide groups determined from graph modeling, rather than protein groups or gene groups, as the quantification unit." This sentence is somewhat difficult to parse. If one uses, say, Percolator, one will get a list of protein accessions from which each identified peptide appears, so these protein groups are essentially peptide groups, which can be used for label free or TMT quantification. Is the major novelty here a categorization scheme to show whether those accessions in a protein group come from the same gene, or the modification of the picked protein method to include shared protein groups and not just unique groups? Ideally this should be more clearly stated.

Response: As clarified in the revised manuscript (see above), the introduction of SEPEP as the quantification unit is the key innovation in this study. Previous methods use parsimonious protein groups, individual genes, individual proteins, or correlated peptides from individual proteins, as the unit of quantification. While using peptides mapping exclusively to a single protein for quantifying that specific protein, as implemented in PQQ and PeCorA, provides accurate quantification, it excludes many peptides that are shared by multiple proteins. On the other hand, when parsimonious protein groups or genes are employed as the quantification units, the isoform-specific information available from shotgun proteomics data is often suppressed or lost. By shifting the quantification unit to SEPEP, i.e., peptides that exclusively map to a group of protein isoforms that are indistinguishable based on the identified peptides, our approach can leverage all confidently identified peptides, including those mapping to multiple proteins or even multiple genes. Moreover, this approach retains and utilizes all the available isoform-distinguishable information present in the data, thus enhancing the detection of possible isoform regulations in our analysis. We hope this information is now clear in the updated Introduction section.

2. Page 5, the authors used an *in silico* analysis (digesting RefSeq protein entries virtually) to estimate the proportion of peptides that may map to one or more isoforms, but how this analysis is done is different from real database search parameters which makes it hard to interpret the results. Only fully tryptic peptides with no miscleavages are used, whereas in a real world scenario the search parameter would include semi-tryptic ends and 1-2 miscleavages allowed which depending on the experiment could account for a substantial portion of peptides. This is especially important for isoform analysis since, as this same group has previously very nicely shown, lysines and arginines are enriched in exon boundaries so often an isoform is discerned by a miscleaved cross-junction peptide. Likewise, peptides of 7 and 8 amino acids should either be removed from consideration or interpreted with care to be in line with community guidelines calling for peptides with at least 9 aa to provide strong evidence of non-canonical sequences. In my opinion this section should either be expanded to match realistic experimental conditions or removed.

Response: Following the reviewer's suggestion, we performed new analyses based on *in silico* digested peptides with 1 and 2 miscleavages and with semi-tryptic peptides. Peptide distributions from these analyses are very similar to that of the fully tryptic peptides with no miscleavage (**Supplementary Fig. 1**). These results are included in the revised manuscript. Moreover, as shown in **Supplementary Table 1**, there were only about 5% identified peptides with 1 or 2 miscleavages in real experiments. We agree with the reviewer that a longer peptide can provide stronger evidence of non-canonical peptides. We would like to clarify that SEPeqQuant is developed primarily to perform global quantification of annotated proteins in a given database, rather than identifying novel peptides or proteins. Keeping peptides with 7 and 8 amino acids is widely used in global protein quantification studies. We will add minimum peptide length as an option in the SEPeqQuant software for users who are interested in non-canonical proteins.

New Text, Page 7:

“Peptide distributions from other *in silico* digestion experiments were similar to that of the experiment with fully digested peptides with no missed cleavages (**Supplementary Fig. 1**).”

Supplementary Fig. 1: Classification of the *in silico* digested peptides with 1 (a) and 2 (b) miscleavages and semi-tryptic peptides (c) based on their mapping to genes and protein isoforms.

3. Page 6, the authors wrote that "Percentages of the single isoform peptides in these studies were slightly higher than that in *in silico* digestion, whereas an opposite trend was found for multi-isoform peptides (Fig. 1d), suggesting competition among multiple isoforms may reduce transcriptional and/or translational efficiency." This statement is somewhat confusing, because the *in silico* digestion treats all proteins as equal, whereas in the experimental data there is a dynamic range of concentration between different proteins. It could just be that proteins with fewer isoforms also tend to have higher natural abundance so would be detected more often (e.g., see the two miscleavage peptide KKEEQEFVWK for NP_001291672.1)

Response: Agreed. We updated the text in the revision and removed the speculative statement.

4. More generally, it would seem that both the *in silico* analysis and the overall inference method are highly dependent on which database is being supplied. Which peptide is partially discriminant or fully isoform discriminant will change based on how many isoforms are in the database. If the database is over-annotated and contains many redundant/unverified isoforms that don't exist in a particular sample then it might stack the odds against unique peptides or razor peptides methods. The RefSeq database being used including NP and XP entries is quite big (~140k sequences) so one would expect there are more class 3 and class 4 SEPEPs than if one were to use UniProt or

a custom database. How that would affect the analysis and results should probably be investigated more thoroughly.

Response: The reviewer is correct that a more comprehensive reference protein database will lead to more class 3 and class 4 SEPEPs. However, for studies aiming to explore protein isoform regulation, a more comprehensive reference protein database such as the RefSeq database is preferred. With around 20K proteins, the UniProt canonical database includes only one canonical sequence for most genes and thus is not suitable for investigating protein isoform regulation. A customized database derived from matched RNASeq data represents the optimal choice, but many proteomic studies do not have matched RNASeq data. We have added these points in the Discussion section in the revised manuscript.

New Text, Page 20:

“To reduce the number of degenerate peptides, a commonly employed strategy is the utilization of the UniProt canonical database. However, due to its limited scope of approximately 20,000 proteins, this database only encompasses a single canonical sequence for most genes, rendering it unsuitable for investigating protein isoform regulation. In our study, we sought to facilitate a comprehensive exploration of protein isoforms by employing the RefSeq database that encompasses both curated proteins (NP and YP) and predicted proteins (XP), resulting in a total of 140,000 entries. When a SEPEP includes both curated and predicted proteins, it makes sense to focus on the curated proteins in further investigation. However, when a SEPEP includes only predicted proteins, it provides direct experimental evidence for the predictions. Opting for a more conserved protein database, such as one exclusively composed of curated proteins, may reclassify certain higher-class SEPEPs into lower classes. However, this approach could potentially overlook regulatory mechanisms involving predicted isoforms. Alternatively, when matched RNASeq data is available, the utilization of customized protein databases derived from such data represents the optimal choice. Notably, SEPEPQuant can also be utilized alongside customized databases.”

5. The title of the manuscript is somewhat ambiguous. It states "comprehensive protein isoform characterization", but of course the isoforms are not necessarily resolved in the method so they are not characterized per se, e.g., a class 3 SEPEP may show some differential regulation but it is not clear which of the actual protein isoform is responsible. A more precise description along the line of recovering potential isoform regulations should be considered.

Response: We thank the reviewer for this comment. The title of the manuscript has been changed to "SEPEPQuant enhances the detection of possible isoform regulations in shotgun proteomics".

6. Page 16: "For genes with multiple SEPEPs, 35.1% - 79.8% had at least one SEPEP with a below 0.5 correlation with the corresponding gene abundance, suggesting extensive isoform-specific regulation." This is an important observation, but should be placed in fuller context

including background variance. What is the range in correlations between individual peptides in the non-discriminative SEPEPs, and is this a function of protein abundance and peptide intensity in the mass spec? To me this would be really important especially for inferring isoforms without any unique peptide whatsoever, e.g., in the example of ACTR3 on page 9, where the short isoform is completely subsumed in the long isoform. Could the differential trends between peptides be more simply explained by post-translational modifications or variance due to low peptide abundance? Is there corroborating evidence at the RNA level?

Response: We thank the reviewer for this insightful comment. Following the suggestion, we performed additional analyses and revised this section based on the new results.

New Text, Page 11:

“Among genes quantified by both methods, the quantifications of the C4 SEPEPs exhibited a strong correlation with their respective host gene quantifications. Only a small fraction of cases, specifically 8% in the iPSC dataset, 1% in the HCC-TMT dataset, and 5% in the HCC-label free dataset, showed correlations below 0.5 (**Fig. 3f**), emphasizing the reliability of SEPEP quantifications. Interestingly, for genes with multiple SEPEPs, a significant number had at least one SEPEP with a correlation less than 0.5 with their host genes, including 65.4%, 35.1%, and 79.8% of genes in the iPSC, HCC-TMT, and HCC-label free data sets, respectively (**Fig. 3f**). To further explore this observation, we assessed the distribution of within-SEPEP peptide correlations and their relationship with average MS1 peptide intensities in the HCC-TMT data set. As expected, peptides with higher MS1 intensities exhibited relatively higher correlations, but the overall correlation remained strong, with a median value of 0.69 (**Supplementary Fig. 2d**). These findings suggest that at least some of the discordance between SEPEP and gene correlations could be attributed to isoform-specific regulation.”

Fig. 3f. Distributions of the correlations between C4 SEPEPs and host genes (black curves) and the lowest correlations between SEPEPs and host genes (pink curves).

Supplementary Fig. 2d. A density plot showing the distribution of within-SEPEP peptide correlations and their relationship with average MS1 peptide intensities.

It's an interesting idea that the differential trends between peptides may also be explained by post-translational modifications. We plan to systematically explore this idea in our further work.

Following the reviewer's suggestion, we also conducted a comparison between the SEPEP and RNASeq isoform results for the examples presented in Figure 4. While the RNASeq data partially support the SEPEP results, they do not entirely align. As an illustration, when examining ACTR3 (see figure below), the parsimonious protein inference assigned all shared peptides to NP_005712.1, suggesting a significant decrease of this isoform in tumors compared to normal adjacent tissues (NATs) ($p=1.9e-4$). SEPEPQuant, on the other hand, reported two SEPEPs, with ACTR3_SEPEP.2_C2 showing a significant increase in tumors compared to NATs ($p=2.3e-11$) and being associated with the three NP_005712.1-specific peptides. The RNASeq data also indicated a significant increase in the transcript isoform (NM_005721.5) encoding NP_005712.1, thus corroborating the SEPEPQuant result.

However, in the proteomic data, both the gene-level quantification (ACTR3 protein group in Fig. 4a and the figure below) and the ACTR3_SEPEP.1_C4 quantification were significantly lower in tumors compared to NATs. Since ACTR3_SEPEP.1_C4 encompasses peptides from both NP_005712.1 and NP_001264069.1, and we have already demonstrated an increase of NP_001264069 in tumors compared to NATs, the possible inference is that NP_005712.1 is decreased in tumors compared to NATs. Surprisingly, in the RNASeq data, both the gene-level quantification (ACTR3 gene) and the isoform-specific quantification for the transcript NM_001277140.1, encoding NP_001264069.1, exhibited significantly higher abundance in tumors compared to NATs. Given that the inconsistency was observed in both the gene-level quantification, which is independent of SEPEP inference, and the SEPEP-based inference for NP_001264069.1, it is unlikely that this discrepancy is solely attributable to a technical issue with SEPEP inference. Alternatively, this could potentially be explained by post-transcriptional regulation, leading to a discrepancy between mRNA and protein results. Similar observations were made for other genes as well.

Considering the challenge of distinguishing biological from technical factors when encountering discrepancies between mRNA and protein results, we decided not to include this comparison in the present paper. As a more straightforward way to validate SEPEP inferences, we have conducted targeted proteomic analysis to confirm the clinically relevant inference regarding an exon skipping event in SLK (see Fig. 6g-h).

Comparison of proteomics and RNASeq results for ACTR3.

Reviewer #2 (Remarks to the Author):

In this manuscript, the authors present SEPEPQuant for SEPEP-level quantification and identification of differentially expressed SEPEPs using bottom-up MS. SEPEPQuant can provide additional information for proteoform quantification, and SEPEPQuant analyses of several data sets demonstrate that it identified proteoform-level regulation events that were missed by protein-level analysis.

Major comments.

1. Several proteogenomic methods have been proposed to analyze splice junction peptides, which are related to SEPEP and should be cited in Section Introduction. For example, Woo et al, Proteogenomic Analysis Reveals Multiple Peptide Mutations and Complex Immunoglobulin Peptides in Colon Cancer, Journal of Proteome Research, 215, 3555-3567.

Response: We thank the reviewer for pointing out this publication. We updated the introduction section to provide background information on this and other related methods.

New Text, Pages 5:

“Leveraging RNASeq data from matched samples, the CustomProDB method constructs a customized protein database with reduced number of protein isoforms by excluding isoforms with low transcript abundance in matched RNASeq data²⁶. This method also enables novel protein isoform identification, but the quantification challenge remains because each gene may still have multiple RNASeq data-supported isoforms. Based on the assumption that each gene only has a dominant isoform, Liu et al.¹⁰ used matched RNASeq data to select one isoform with the highest transcript abundance for each gene. Woo et al.²⁷ and Lau et al.²⁸ focus on novel protein isoforms by identifying and quantifying peptides mapped to novel isoform junctions detected based on RNASeq data. While these methods are appealing, their utility in the majority of proteomic studies is limited by the prerequisite of matched RNASeq data.”

2. Parsimony-based protein inference can be easily extended to parsimony-based proteoform inference. The difference between SEPEP and parsimony-based proteoform inference should be clarified and discussed. In Fig 3d, the SEPEP approach needs to be compared with parsimony-based protein inference and proteoform inference.

Response: We thank the reviewer for this comment, but we are not sure whether we fully understand the question. Based on our understanding, proteoforms of a gene may arise from transcript isoforms and site specific alterations including both PTMs and SNVs (<https://www.nature.com/articles/nchembio.2576>). The current version of SEPEPQuant is designed for the analysis of global proteomics data without considering PTMs and SNVs, therefore, parsimony-based proteoform inference in our study is equivalent to parsimony-based protein isoform inference. During the revision, we further applied the parsimony-based protein isoform inference in MaxQuant to the iPSC dataset (**Supplementary Fig. 2c**), and the results are consistent with our previous analysis based on the FragPipe implementation (**Fig. 3d**). SEPEPQuant reported more genes with multiple features than both FragPipe and MaxQuant. This has been included in the revised manuscript.

New Text, Page 10:

“Upon conducting an additional comparison between SEPEPQuant and MaxQuant using the iPSC dataset, we obtained similar results (**Supplementary Fig. 2c**).”

		iPSC-MaxQuant	
		SEPEP-based	Protein group
	Gene with multiple features	861	20
Feature distribution	2	638	18
	3	170	1
	5	33	0
	5	13	0
	>=6	20	1

Supplementary Fig 2c. Comparison of numbers of genes with multiple SEPEPs in SEPEPQuant analysis and those with multiple protein groups in MaxQuant analysis.

3. In SEPEP FDR control, the sentence “Specifically, a SEPEP is considered a target hit if the highest scoring peptide in the SEPEP is from a forward protein sequence.” is confusing. All peptides in a SEPEP are structurally equivalent. If the highest-scoring peptide is from target proteoforms, all other peptides in the same SEPEP should be from the same target proteoforms. Is it possible that a peptide is shared by target and decoy proteoforms. In this case, how to determine if the SEPEP is a target or decoy identification.

Response: Sorry about the confusion, and the reviewer’s interpretation is correct. We have updated the text to correct this. If a peptide is shared by both target and decoy proteoforms, we treat the peptide as a target peptide because the prior probability is higher for the peptide to be associated with a target.

New Text, Page 9:

“Specifically, a SEPEP is considered a target hit if the peptides in the SEPEP are from a forward protein sequence, and a decoy hit if the peptides are from a decoy protein sequence.”

4. In TMT data, the peptides of a SEPEP identified from one sample may be different from those of the same SEPEP identified from another sample. When tens of samples are analyzed, the problem becomes more complicated. It is unclear how to do SEPEP level quantification for TMT data.

Response: The method for SEPEP level quantification is equivalent to routinely used methods for gene level or protein group level quantification. For TMT data, peptide identification is performed for each TMT plex, whereas quantification for each sample is based on the reporter ion intensities of each sample as compared to a common reference sample that is included in each TMT for normalization purpose, i.e., experimental sample to reference sample ratio. For a SEPEP, as described in the Methods section, the median value of the TMT ratios of all peptides belonging to the SEPEP is used to quantify the SEPEP.

5. Fig 4 shows several examples of differentially expressed SEPEPs. The iPSC and HCC data sets contain both RNA-Seq and MS data. Can the SEPEP abundances of the isoforms in Fig. 4 be compared with their transcript expression levels?

Response: We appreciate the reviewer's suggestion and have conducted a comparison between the SEPEP and RNASeq isoform results for the examples presented in Figure 4. While the RNASeq data partially support the SEPEP results, they do not entirely align. As an illustration, when examining ACTR3 (see figure below), the parsimonious protein inference assigned all shared peptides to NP_005712.1, suggesting a significant decrease of this isoform in tumors compared to normal adjacent tissues (NATs) ($p=1.9e-4$). SEPEPQuant, on the other hand, reported two SEPEPs, with ACTR3_SEPEP.2_C2 showing a significant increase in tumors compared to NATs ($p=2.3e-11$) and being associated with the three NP_005712.1-specific peptides. The RNASeq data also indicated a significant increase in the transcript isoform (NM_005721.5) encoding NP_005712.1, thus corroborating the SEPEP result.

However, in the proteomic data, both the gene-level quantification (ACTR3 protein group in Fig. 4a and the figure below) and the ACTR3_SEPEP.1_C4 quantification were significantly lower in tumors compared to NATs. Since ACTR3_SEPEP.1_C4 encompasses peptides from both NP_005712.1 and NP_001264069.1, and we have already demonstrated an increase of NP_001264069 in tumors compared to NATs, the possible inference is that NP_005712.1 is decreased in tumors compared to NATs. Surprisingly, in the RNASeq data, both the gene-level quantification (ACTR3 gene) and the isoform-specific quantification for the transcript NM_001277140.1, encoding NP_001264069.1, exhibited significantly higher abundance in tumors compared to NATs. Given that the inconsistency was observed in both the gene-level quantification, which is independent of SEPEP inference, and the SEPEP-based inference for NP_001264069.1, it is unlikely that this discrepancy is solely attributable to a technical issue with SEPEP inference. Alternatively, this could potentially be explained by post-transcriptional regulation, leading to a discrepancy between mRNA and protein results. Similar observations were made for other genes as well.

Considering the challenge of distinguishing biological from technical factors when encountering discrepancies between mRNA and protein results, we have decided not to include this comparison in the present paper. As a more straightforward way to validate SEPEP inferences, we have conducted targeted proteomic analysis to confirm the clinically relevant inference regarding an exon skipping event in SLK (see **Fig. 6g-h**).

Comparison of proteomics and RNASeq results for ACTR3.

Minor comments

1. In Fig. 2b, protein information in the tripartite graph is not needed to identify SEPEPs, which should be pointed out in the manuscript.

Response: This is correct and has been clarified in the revised manuscript.

New Text, Page 9:

“Of note, the gene vertices do not affect the identification of SEPEPs, but they help organize protein vertices into genes and classify SEPEPs into single gene or multi-gene SEPEPs (see below) to facilitate data interpretation”.

2. The word “group” is not included in SEPEP, which makes it unclear if a SEPEP is a peptide or a peptide group. The authors may consider changing SEPEP to include the word “group.”

Response: Thank you for bringing attention to this potentially confusing aspect. This has been clarified in the revised manuscript.

New Text, Page 9:

“To clarify, the term SEPEP is used to denote a specific grouping of peptides that exhibit structural equivalence within the context of the tripartite graph instead of an individual peptide.”

3. For Fig 3e, the authors gave an explanation: “more stringent FDR control caused by the much larger number of candidate SEPEPs compared with candidate genes.” The explanation might be incorrect. The reason might be that more stringent FDR control was caused by the less number of candidate SEPEPs compared with the candidate peptides.

Response: We agree that the FDR explanation is somewhat speculative. We provided an alternative, more confident explanation in the revised manuscript.

New Text, Page 10-11:

“We further compared genes harboring quantifiable SEPEPs with quantifiable genes reported based on parsimonious inference (**Fig. 3e**). The numbers of genes harboring quantifiable SEPEPs were smaller than corresponding gene numbers reported by parsimonious inference across all three data sets. However, SEPEPQuant also reported 1,318, 3,122, and 975 multi-gene SEPEPs for the three data sets, respectively, and such information is missing or difficult to track in existing computational tools (**Fig. 3c**).”

4. The resolution of Fig. 5a, Fig. 6a, Fig.6b is low.

Response: The resolution of these figures has been improved in the revised manuscript.

Reviewer #3 (Remarks to the Author):

The article ‘SEPEPQuant enables comprehensive protein isoform characterization in shotgun proteomics’ by Dou et al. introduces a new methodology to characterize isoforms from shotgun proteomics data. The method addresses a very important topic, as isoforms of many proteins have been shown to regulate biological processes differently – sometimes in opposing manners – yet many proteomic studies ignore isoforms due to a lack of suitable methods to distinguish isoforms.

The authors propose a new graph-based method, where they represent the relations between genes, proteins and peptides through a tripartite graph. This relatively simple representation allows the authors to identify isoforms of proteins not possible with a parsimonious method to which the authors compare their approach. The paper has a big potential to advance the field and help experimental researchers to identify protein isoforms, which may have a big impact on the investigated biological system.

The biggest limitation, in my opinion, is the current lack of independent verification that their method works. For example, an independent experiment could be performed to confirm the presence and quantity of the identified isoforms as predicted by the author's method. For example, Western Blots of isoforms identified could be performed to confirm that the methods the authors propose give compatible results with the Western Blots (if suitable antibodies are available for the proteins discussed in the manuscript. However, the method was applied to big datasets so surely some of those should allow for independent measurements with Western Blot or otherwise). Besides this major comment, I include several smaller comments below that should be addressed before the manuscript can be considered for publication.

Response: We thank the reviewer for recognizing the potential impact of our method. We also appreciate the great suggestion on independent verification of our findings. As a computational lab, it was difficult for us to conduct new experimental validation for the iPSC study. However, we successfully collaborated with the authors of the HCC study and conducted a targeted parallel reaction monitoring (PRM) experiment to verify the correlation between the exon skipping event in SLK and tumor initiation and prognosis in HCC. We believe that the inclusion of this new experiment has significantly enhanced the quality and impact of our manuscript. Please refer to our response to the comment at the end for detailed results from the experiment.

The authors compare their approach to 'the parsimonious' approach. But as they say, there are different implementations by different groups. How much difference do those published methods have in terms of isoform characterisation? Why did the authors only pick FragPipe as a comparison, and not other tools? A more systematic comparison of the different tools available for at least some of the data presented would strengthen the present paper.

Response: Thanks for the comment. We further compared SEPepQuant with MaxQuant using the iPSC dataset and included the results in the revised manuscript.

New Text, Page 10:

“Upon conducting an additional comparison between SEPepQuant and MaxQuant using the iPSC dataset, we obtained similar results (**Supplementary Fig. 2c**).”

		iPSC-MaxQuant	
		SEPEP-based	Protein group
	Gene with multiple features	861	20
Feature distribution	2	638	18
	3	170	1
	5	33	0
	5	13	0
	>=6	20	1

Supplementary Fig 2c. Comparison of numbers of genes with multiple SEPEPs in SEPepQuant analysis and those with multiple protein groups in MaxQuant analysis.

Abstract: protein isoform characterization is inhibited: inhibited sounds a bit strong. Maybe: ... is challenging due to the extensive ...

Response: Agreed, and this statement has been updated in the revised manuscript.

New Text, Page 2:

“protein isoform characterization is challenging due to the extensive number of peptides shared across proteins”.

p 3: the extent to which transcript isoform complexity propagates to the proteome remains controversial: This sentence seems to say that not all transcript isoform complexity may translate to protein complexity, i.e. complexity is reduced. This can happen, but there can also be higher complexity on the protein level due to PTM etc.

Response: The reviewer is correct, isoform complexity may be reduced from transcript isoforms to protein isoforms, but new complexities will be introduced at the PTM level. This study focuses only on isoform complexity. Both SEPepQuant and the global proteomics datasets used in this study are not suitable for exploring the full proteoform complexity.

p6 This could be explained by possibly higher abundance of these peptides because they can be derived from multiple genes: I do not fully understand this explanation. If a peptide is derived from multiple genes, why is this not reflected in the in silico study? The RefSeq protein database should then also include these peptides multiple times. I do not understand why this explains a mismatch between in silico experiments and real data.

Response: Sorry about the confusion. This has been clarified in the revised manuscript.

New Text, Page 7-8:

“This observation may be explained by the higher likelihood of detecting these peptides in data-dependent MS experiments because they can be derived from multiple genes.”

Fig 1 d, e: the caption should briefly mention what we see. In general, most captions in this paper could be a bit more detailed

Response: We thank the reviewer for this comment. We have updated captions of Fig 1d and 1e in the revised manuscript. Improvements have also been made for other figure captions.

New Text, Page 27:

“**d** Classification of the experimentally identified peptides in individual samples in an iPSC cell line study and two hepatocellular carcinoma (HCC) studies into three categories, equivalent to the pie chart on the left in **c**. **e** Further classification of peptides in the multi-isoform group in **d** into three categories, equivalent to the pie chart on the right in **c**.”

P6, middle: in consistent: remove: in

Response: updated.

New Text, Page 8:

“experimental data are largely consistent with the *in silico* digestion results.”

P8 middle: the 50% threshold for missing data: how robust is the method when that threshold is changed? Also is the same threshold applied to the parsimonious approach?

Response: The 50% threshold for missing data was used for both SEPEPQuant and the parsimonious approaches. We obtained the same conclusions using different thresholds, but 50% was selected for publication because this threshold is widely used in published proteomics studies.

Fig 3f: state exactly over what data the correlations are calculated (in the text or supplement)

Response: This has been clarified in the revised manuscript.

New Text, Page 11:

“Among genes quantified by both methods, the quantifications of the C4 SEPEPs exhibited a strong correlation with their respective host gene quantifications. Only a small fraction of cases,

specifically 8% in the iPSC dataset, 1% in the HCC-TMT dataset, and 5% in the HCC-label free dataset, showed correlations below 0.5 (**Fig. 3f**), emphasizing the reliability of SEPEP quantifications. Interestingly, for genes with multiple SEPEPs, a significant number had at least one SEPEP with a correlation less than 0.5 with their host genes, including 65.4%, 35.1%, and 79.8% of genes in the iPSC, HCC-TMT, and HCC-label free data sets, respectively (**Fig. 3f**)."

Fig 4: the choice of investigated proteins should be motivated, as otherwise, one could get the impression that the authors cherry-picked proteins with a big difference in their method to the parsimonious approach. Tables showcasing all proteins should also be presented in the supplement

Response: These genes were selected as representative examples to illustrate different types of drawbacks of parsimony-based protein inference. This has been clarified in the revised manuscript. Comprehensive results for this dataset is reported in the section "**Protein isoforms associated with liver cancer development and prognosis**".

New Text, Page 11:

"Here we selected representative examples from the HCC-TMT data set to illustrate the drawback of these simplifications on protein isoform characterization, and the effectiveness of SEPepQuant in addressing these limitations."

Fig 4a: I do not fully understand if the statement in the text 'whereas the parsimonious inference provided only incorrect information for the long isoform.' is trivial? Is this solely based on the fact that the parsimonious approach assigns every peptide to the long isoform, so naturally that information is incorrect as it also includes all peptides from the short isoform? If this is the case, I would not say that this approach shows anything incorrectly about the long isoform. This approach does not show the long isoform at all! It shows data about a group of proteins, one cannot really say whether this is the long or short isoform.

Response: It is true that the parsimonious approach reports data for a group of proteins instead of just the long isoform. However, in reality, for downstream analysis requiring a single protein, the long isoform is selected by default for reporting, which leads to incorrect information. We have updated our statement in the revised manuscript to clarify our point.

New Text, Page 12:

"Thus, SEPepQuant provided useful information for both isoforms, whereas the parsimonious inference assigned all peptides to the long isoform, leading to inaccurate quantification because some peptides may actually belong to the short isoform despite being mappable to both isoforms."

Fig 5/Supp Fig 2/ p12: The investigation of correlations of different isoforms with time is very promising. However, despite statements such as 'these observations still revealed important roles

of these gene families in iPSC differentiation', the section contains few ideas of what we could learn about the biology of stem cell differentiation from these findings. While this is not the major objective of the paper, I believe the method could gain more acceptance if some striking biological interpretations could be obtained. More importantly, independent experiments should confirm the new findings

Response: We thank the reviewer for this important comment. As mentioned in our response to the overall comment, it was difficult for us to conduct new experimental validation for the iPSC study. To confirm our new finding on DPYSL3, we leveraged matched RNAseq data from the original publication and included the results in the revised manuscript.

New Text, Page 15:

“This observation was further validated by the RNASeq data obtained from the original publication, despite the limited number of time points profiled (four in total). Specifically, the transcript NM_001197294.2, which corresponds to NP_001184223.1, showed a consistent downward trend from day 0 to day 14, whereas the other transcripts and the gene level measurements exhibited the lowest values at day 7 but showed an increase by day 14 (Supplementary Fig. 3f).”

Supplementary Fig. 3f. Gene and transcript isoform expression of DPYSL3 in RNASeq data.

In addition, we collaborated with the authors of the HCC study and conducted a targeted parallel reaction monitoring (PRM) experiment to verify the correlation between the exon skipping event in SLK and its impact on tumor initiation and prognosis in HCC. This was also included in the revised manuscript.

New Text, Page 17:

“In order to verify the correlation between the exon skipping event in SLK and tumor initiation and prognosis in HCC, we performed parallel reaction monitoring (PRM) analysis on 20 paired tumor and NAT samples selected from the HCC-TMT study (**Supplementary Table 5**). Among these sample pairs, 10 were obtained from patients who passed away within 12 months of tissue collection (poor prognosis), while the other 10 pairs were obtained from patients who survived for more than 40 months after tissue collection (good prognosis). For the PRM experiment, we specifically chose five peptides that had been previously identified in the TMT study (**Supplementary Table 5**). One of these peptides, KKEEQEFVQK, was found exclusively in the isoform NP_001291672.1, which resulted from exon skipping. Our analysis revealed a significant increase in the abundance of this peptide in tumor samples compared to NATs (**Fig. 6g**). Moreover, when comparing poor prognosis tumors to good prognosis tumors, we observed a higher abundance of KKEEQEFVQK in the former (**Fig. 6h**). Conversely, another peptide EVINEVEK, which exclusively mapped to the two exon inclusion isoforms, displayed the opposite pattern. Its abundance was decreased in tumors compared to NATs, and further decreased in poor prognosis tumors compared to good prognosis tumors (**Fig. 6g-h**). The other three peptides, shared by both the exon skipping and exon inclusion isoforms, demonstrated either lower levels of increase or even a decrease in abundance when comparing tumor samples to normal samples, as well as when comparing poor prognosis tumors to good prognosis tumors (**Fig. 6g-h**). These results provide robust evidence confirming the association between the exon skipping event in SLK and tumor initiation and prognosis in HCC.”

Table s5b: TMT data for selected samples and SLK peptides for PRM validation.

peptide	Protein list	<12 month										>40 month										DE		Survival	
		T313	T351	T380	T919	T487	T1021	T949	T1019	T517	T385	T487	T513	T227	T399	T271	T225	T389	T485	T207	T399	P value	Log2FC	P value	HR
time (month)	NA	20	10	8	3	3	3	3	3	3	3	3	3	3	3	3	3	3	3	3	3	NA	NA	NA	NA
status	NA	Dead	Dead	Dead	Dead	Dead	Dead	Dead	Dead	Dead	Dead	Living	Living	Living	Living	Living	Living	Living	Living	Living	Living	NA	NA	NA	NA
KKEEQEFVQK	NP_001291672.1	1.39025	1.16233	1.09021	0.87295	0.68979	0.66191	0.58995	0.53289	0.4054	0.39959	-0.1946	-0.2051	-0.2111	-0.2555	-0.2726	-0.3056	-0.4179	-0.6208	-0.6481	-1.2312	1.5E-07	1.81878	0.03867	2.58
DTILQTVDLVSQETGEK	NP_001291672.1, NP_055335.2	0.30812	-0.1051	0.14481	0.07793	-0.2125	-0.0992	0.18033	-0.3719	0.08651	-0.1778	-0.6029	-0.5189	0.68836	NA	0.55552	0.51177	NA	0.0505	-0.6229	NA	0.95486	-0.0115	0.067	3.84
ILNEKPTTDEPEK	NP_001291672.1, NP_055335.2	0.4059	NA	0.26788	0.04837	0.24554	0.24484	0.17406	-0.0363	0.54695	-0.4909	-0.1424	-0.2183	0.71992	-0.8515	0.49188	0.27141	-0.2399	0.0605	NA	-0.413	0.3384	0.17504	0.181	1.46
ELDEEHSQELK	NP_001291672.1, NP_055335.2, NP_011338705	0.2088	-0.2171	0.71393	0.39897	0.3435	-0.0003	-0.0106	-0.1400	0.02193	-0.0559	-0.0343	-0.3951	0.18521	-0.0418	0.43798	0.0976	-0.1918	0.1965	-0.618	0.5001	0.1085	0.27482	0.0750	3.81
EVINEVEK	NP_055335.2, ILK, NP_011338705.1, ILK	NA	-0.4411	NA	NA	NA	0.09784	-0.8557	NA	NA	NA	NA	NA	NA	-0.8621	1.18737	0.30365	-0.0555	0.30005	-0.3883	-0.3173	0.09647	-1.1404	0.18	0.62
Matched normal																									
peptide	Protein list	P213	P362	P396	P816	P498	P1022	P388	P1018	P518	P286	P488	P518	P228	P396	P272	P222	P386	P386	P358	P394	P value	Log2FC	P value	HR
KKEEQEFVQK	NP_001291672.1	-0.0197	0.00834	0.09275	0.02973	0.53828	-0.1638	-0.0935	-0.0559	-0.2785	-0.37	-0.0874	-0.1931	-0.1621	-0.4464	-0.1322	-0.4493	-0.4294	-0.3998	-0.7252	-0.818	0.00136	0.27682	NA	NA
DTILQTVDLVSQETGEK	NP_001291672.1, NP_055335.2	-0.2705	-0.2051	0.18195	-0.1184	0.18175	-0.2326	0.11435	-0.3284	-0.1481	0.44519	-0.0913	-0.2563	-0.0018	NA	-0.2080	0.11839	NA	-0.2215	-0.0475	NA	0.65651	-0.042	NA	NA
ILNEKPTTDEPEK	NP_001291672.1, NP_055335.2	0.12078	NA	0.4157	0.14679	0.30305	0.27194	0.2742	0.07728	0.28961	0.49954	0.29067	0.02874	0.20931	0.18985	0.05608	-0.157	0.18075	0.1489	NA	0.0454	0.02388	0.1232	NA	NA
ELDEEHSQELK	NP_001291672.1, NP_055335.2, NP_011338705	-0.1468	-0.493	0.05133	0.30923	-0.1358	-0.219	-0.0888	-0.0869	0.37884	-0.0296	-0.0848	-0.0992	-0.0943	-0.1305	-0.1351	-0.1081	-0.0218	-0.3194	-0.1099	-0.2113	0.29279	0.07903	NA	NA
EVINEVEK	NP_055335.2, ILK, NP_011338705.1, ILK	NA	0.05125	NA	NA	NA	-0.38407	0.05847	NA	NA	NA	NA	NA	NA	NA	-0.3586	0.89136	-0.3506	-0.1881	0.06605	0.31122	-0.3723	0.12787	0.5668	NA

Figure 6g: PRM peptide abundance comparison between tumor and matched normal of selected tumor and normal pairs.

Figure 6h: PRM peptide abundance comparison between tumors from patients with good or poor prognosis.

5a: the authors identified gene/SEPEP pairs that are not correlated in tissue time (and similar analysis for tumour versus non-tumour in 6a). Did the authors perform some correction for multiple testing to rule out that with so many genes investigated, it did not occur just by chance that some parts are anticorrelated?

Response: We thank the reviewer for this important comment. We applied multiple test adjustment for these figures and updated our results and figures in the revised manuscript. As shown below, despite the reduced numbers of significant SEPEPs, our conclusions remain the same after the adjustment.

Figure 5a: Scatter plot comparing time-dependent regulations at gene and SEPEP levels.

Figure 6a: Scatter plot of tumor versus normal comparison results at gene and SEPEP levels.

Reviewer #1 (Remarks to the Author):

Thank you for the thoughtful responses. I have no further comments.

Reviewer #2 (Remarks to the Author):

The authors have addressed all my concerns in the revised version. The revised version of the manuscript can be accepted for publication.

Reviewer #3 (Remarks to the Author):

The authors have addressed all my comments well, apart from a minor point about how they present their data below. I can now recommend the paper for publication.

Supplementary Figure 2c: the presentation is a bit confusing, and I needed to jump back and forth between the text and main fig 3d to (hopefully) understand what the authors have done. Contrary to the caption in supp fig 2c, this table does not seem to show a comparison of SEPepQuant and MaxQuant, but just shows the results from MaxQuant, while the results of SEPepQuant are in main fig 3d. Then, it is a bit hard to compare supp fig 2c and fig 3d because they are neither side by side nor normalised. It may be best to show some normalised numbers side by side in the supplement.

There is also a typo: two 5's are the feature distribution.

Fig 3f: typo: sorrelation

Re: NCOMMS-22-41664A “SEPEPQuant enhances the detection of possible isoform regulations in shotgun proteomics”

REVISIONS IN RESPONSE TO REVIEWERS' COMMENTS

We thank the reviewers for insightful comments that helped improve our manuscript. We have considered all comments and suggestions and revised the manuscript accordingly. Please see below for a point-by-point response to each of the points made by the reviewers. For your convenience, we have highlighted the major changes in green in the revised manuscript. Reviewers' comments, our response, and new text in the revised manuscript are colored in black, blue, and green, respectively, in this document.

REVIEWER COMMENTS

Reviewer #1 (Remarks to the Author):

Thank you for the thoughtful responses. I have no further comments.

Reviewer #2 (Remarks to the Author):

The authors have addressed all my concerns in the revised version. The revised version of the manuscript can be accepted for publication.

Reviewer #3 (Remarks to the Author):

The authors have addressed all my comments well, apart from a minor point about how they present their data below. I can now recommend the paper for publication.

Supplementary Figure 2c: the presentation is a bit confusing, and I needed to jump back and forth between the text and main fig 3d to (hopefully) understand what the authors have done. Contrary to the caption in supp fig 2c, this table does not seem to show a comparison of SEPEPQuant and MaxQuant, but just shows the results from MaxQuant, while the results of SEPEPQuant are in main fig 3d. Then, it is a bit hard to compare supp fig 2c and fig 3d because they are neither side by side nor normalised. It may be best to show some normalised numbers side by side in the supplement.

Response: We are sorry about the confusion. SEPEPQuant is a post-processing tool that functions with peptide-level outputs from search engines such as FragPipe or MaxQuant. Supplementary Figure 2c includes results from MaxQuant either with (the SEPEP-based column) or without (the Protein group column) post-processing by SEPEPQuant. From this table, it is clear that SEPEP-based post-processing greatly increased the number of genes with multiple features (861 vs 20). This is in the same format as the FragPipe-based results presented in Figure 3d.

We have revised our statement to eliminate this confusion.

New Text, Page 10-11:

“To assess the broader applicability of this finding, we performed a supplementary analysis on the iPSC dataset using MaxQuant for database searching. Similarly, compared with the protein group level quantification reported by MaxQuant, SEPEPQuant identified 43 time more genes with multiple features (**Supplementary Fig. 2c**).”

There is also a typo: two 5's are the feature distribution.

Fig 3f: typo: sorrelation

Response: Thanks for pointing these out. We have fixed these issues in the revised manuscript.